# Two-stage binding of mitochondrial ferredoxin-2 to the core iron-sulfur cluster assembly complex

Ralf Steinhilper [1], Linda Boß[2,3], Sven-A. Freibert[2,3], Vinzent Schulz[2,3], Nils Krapoth[2,3], Susann Kaltwasser [4], Roland Lill [2,3] ✉ & Bonnie J. Murphy [1] ✉

Iron-sulfur (FeS) protein biogenesis in eukaryotes begins with the de novo assembly of [2Fe-2S] clusters by the mitochondrial core iron-sulfur cluster assembly (ISC) complex. This complex comprises the scaffold protein ISCU2, the cysteine desulfurase subcomplex NFS1-ISD11-ACP1, the allosteric activator frataxin (FXN) and the electron donor ferredoxin-2 (FDX2). The structural interaction of FDX2 with the complex remains unclear. Here, we present cryo-EM structures of the human FDX2-bound core ISC complex showing that FDX2 and FXN compete for overlapping binding sites. FDX2 binds in either a 'distal' conformation, where its helix F interacts electrostatically with an arginine patch of NFS1, or a 'proximal' conformation, where this interaction tightens and the FDX2-specific C terminus binds to NFS1, facilitating the movement of the [2Fe-2S] cluster of FDX2 closer to the ISCU2 FeS cluster assembly site for rapid electron transfer. Structure-based mutational studies verify the contact areas of FDX2 within the core ISC complex.

Iron-sulfur (FeS) clusters are ancient metallocofactors that are essential for numerous cellular processes, including oxidative phosphorylation, the citric acid cycle, and DNA replication and repair. In addition to their most prominent role in electron transfer and storage, FeS clusters can also be directly involved in catalysis or mediate structural support[1,2]. The biosynthesis of FeS clusters is carried out by protein-based machineries present in almost all living organisms[3]. In eukaryotes, the process involves the mitochondrial iron-sulfur cluster assembly (ISC) machinery and the cytosolic iron-sulfur protein assembly (CIA) system, which catalyze the biogenesis of both [2Fe-2S] and [4Fe-4S] proteins in mitochondria, cytosol, and nucleus.

The process is initiated by the mitochondrial core ISC complex, which synthesizes a [2Fe-2S] cluster de novo from cysteine-derived sulfur and ferrous iron ($Fe^{2+}$)[4,5]. This complex is composed of the iron-sulfur scaffold protein ISCU2, the dimeric pyridoxal phosphate (PLP)-dependent cysteine desulfurase subcomplex NFS1-ISD11-ACP1, the

allosteric regulator frataxin (FXN), and the electron donor ferredoxin-2 (FDX2), which is itself reduced by the NADPH-dependent ferredoxin reductase (FDXR)[6–11]. The reaction begins with the desulfuration of free L-cysteine at the PLP cofactor of NFS1 to produce L-alanine and a persulfide (-SSH) at residue Cys381[NFS1] (ref. 12). The strictly conserved Cys381 is located on a flexible loop of NFS1 (termed 'Cys-loop') allowing the movement of the persulfide over a distance of more than 20 Å to the FeS cluster assembly site of ISCU2, which binds $Fe^{2+}$ (ref. 13,14). Persulfide transfer from Cys381[NFS1] to Cys138[ISCU2], one of three conserved cysteine residues of ISCU2, is facilitated by FXN, which was shown to increase persulfide transfer rates[15–17] by changing the local environment of the ISCU2 assembly site[18,19]. The sulfane sulfur ($S^0$) of the persulfide on ISCU2 is reduced to sulfide ($S^{2-}$) by FDX2 to initiate [2Fe-2S] cluster formation[10]. Recent in vitro studies have shown that probably one electron is provided by FDX2, whereas a second electron could be derived from the oxidation of ISCU2-bound $Fe^{2+}$ to

[1]Redox and Metalloprotein Research Group, Max Planck Institute of Biophysics, Max-von-Laue-Str. 3, 60438 Frankfurt am Main, Germany. [2]Institut für Zytobiologie, Philipps-Universität Marburg, Karl-von-Frisch-Str. 14, 35032 Marburg, Germany. [3]Zentrum für Synthetische Mikrobiologie Synmikro, Karl-von-Frisch-Str. 14, 35032 Marburg, Germany. [4]Central Electron Microscopy Facility, Max Planck Institute of Biophysics, Max-von-Laue-Str. 3, 60438 Frankfurt am Main, Germany. ✉e-mail: lill@staff.uni-marburg.de; bonnie.murphy@biophys.mpg.de

ferric iron (Fe$^{3+}$)[14]. Tyr35$^{ISCU2}$-induced dimerization of two ISCU2 proteins, assumed to each contribute a [1Fe-1S] moiety, is required to form a [2Fe-2S] cluster on ISCU2[20–22]. [2Fe-2S] units, generated by the core ISC complex, can be inserted into recipient proteins or serve as building blocks for the assembly of [4Fe-4S] clusters by components of the late ISC machinery[4,23].

Mutations in genes encoding components of the core ISC complex are associated with iron accumulation in mitochondria and have been linked to several rare human diseases[24]. Friedreich's ataxia, the most common dysfunction of FeS cluster biosynthesis, is caused by decreased function of FXN in mitochondria, resulting in neuronal degeneration[25]. Different pathologies arising from mutations in the FDX2 and FDXR genes have been reported[26–28]. Understanding the interplay of FXN and FDX2 with the core ISC complex on the molecular level will contribute to a fundamental understanding of the essential process of FeS cluster biosynthesis and may contribute to efforts at treating diseases caused by aberrant FeS cluster formation.

Despite several attempts to map ferredoxin binding to the core ISC complex, the precise interaction remains uncertain. Competitive binding of bacterial frataxin (CyaY) and ferredoxin (Fdx) on the IscS dimer was proposed from nuclear magnetic resonance (NMR) spectroscopy[29,30], and biochemical experiments[31] suggested that the yeast ISC proteins bind in a sequential and transient fashion. In contrast, a model based on NMR data[10] (recording the interaction of yeast ferredoxin (Yah1) with Isu1) and small-angle X-ray scattering (SAXS) envelopes of *Chaetomium thermophilum* homologues of the ISC machinery suggested a simultaneous binding of frataxin and ferredoxin to distinct sites[13]. Humans express two ferredoxin isoforms, FDX1 (aka adrenodoxin) and FDX2[32]. FDX1 is most abundant in adrenal gland tissues, where it is involved in steroidogenesis, and was recently shown to be essential for the formation of the lipoyl and heme *a* cofactors in all cell types[11]. FDX2 is ubiquitously expressed and is crucial for both [2Fe-2S] and [4Fe-4S] cluster biosynthesis[8]. The C-terminal residues of the two human isoforms play a decisive role in defining their target specificity[11].

Previous X-ray crystallography and electron cryo-microscopy (cryo-EM) structures delivered important information on the overall architecture of the human core ISC complex[13] and its mode of interaction with FXN[18,19]. However, the earlier studies[13,18] were performed in an aerobic atmosphere with copurified zinc (Zn$^{2+}$) in the ISCU2 assembly site[33]. The presence of Zn$^{2+}$ is incompatible with the physiological [2Fe-2S] cluster formation because persulfide reduction by FDX2 is precluded in Zn$^{2+}$-bound ISCU2[14].

In this study, we present high-resolution cryo-EM structures of the human Fe$^{2+}$-containing FDX2-bound core ISC complex by preparing grids under anaerobic conditions employing NADPH-FDXR-reduced FDX2. The structures show that FDX2 binding can occur in two related conformations, only one of which is likely to allow efficient electron transfer. FDX2 and FXN compete for binding to overlapping sites at the core ISC complex during de novo [2Fe-2S] cluster biosynthesis. The contact areas of FDX2 within the core ISC complex were examined by generating FDX2 and NFS1 mutant proteins and testing them in a functional assay of de novo [2Fe-2S] cluster biosynthesis. Our results considerably advance the mechanistic knowledge of FDX2 function in the first step of mitochondrial FeS protein biogenesis.

## Results

### Structure determination of FDX2-bound core ISC complexes under anaerobic conditions

The first aim of this study was to obtain cryo-EM structures of the biosynthetic NFS1-ISD11-ACP1-ISCU2 (hereafter abbreviated (NIAU)$_2$) complex with bound FDX2 and FXN. To mimic near-physiological conditions for de novo FeS cluster biosynthesis during structure determination, we prepared and vitrified all cryo-EM samples under anaerobic conditions. FDX2 was reduced by catalytic amounts of FDXR

and 1 mM NADPH in the buffer, because in this form yeast and human ferredoxins were shown to bind more tightly to the scaffold protein Isu1[10]. Cryo-EM samples were prepared under two different conditions (for details see Methods). First, to elucidate the interaction site of reduced FDX2 with the (NIAU)$_2$ complex, the (NIA)$_2$ subcomplex was incubated for 20 min with iron-loaded ISCU2 and FDXR-reduced FDX2 to yield the (NIAUF)$_2$ sample. Second, a sample was prepared in exactly the same manner, but was incubated on ice and contained equimolar amounts of FXN and reduced FDX2 (hence abbreviated (NIAUXF)$_2$). Cysteine was added ('turnover' conditions), and the sample was quickly vitrified, resulting in an incubation time of less than 1 min at ≤ 4 °C. This condition allows persulfidation of ISCU2 but only slow [2Fe-2S] cluster synthesis[19].

Structures of samples from both conditions were determined by single-particle cryo-EM, reaching resolutions of 2.0 Å for the (NIAUF)$_2$ and 2.1 Å for the (NIAUXF)$_2$ turnover conditions, respectively (Supplementary Figs. 1 and 2). Up to the consensus refinement step the overall architecture of the complexes appeared similar for the two conditions, showing a symmetric (NIAU)$_2$ dimer with FDX2 tightly bound in a cavity between NFS1/NFS1' and ISCU2 (Fig. 1). Additional weak density at this site (Supplementary Fig. 3), typically associated with conformational and/or compositional heterogeneity in cryo-EM, was further assessed by C2 symmetry expansion and focused 3D classification (Supplementary Figs. 4 and 5). This revealed two major FDX2 conformations that we refer to as 'proximal' and 'distal' according to the distances between the [2Fe-2S] cluster of FDX2 and the FeS cluster assembly site of ISCU2, namely 23 Å for the distal conformation and 14 Å for the proximal conformation (Fig. 2a,b). In the (NIAUF)$_2$ dataset, 61% of the particles belong to a class in which FDX2 is bound in the proximal conformation, and 39% belong to a distal-bound FDX2 class (Supplementary Fig. 4). For the turnover dataset, heterogeneity at the FDX2 binding site could be separated into three major classes (Supplementary Fig. 5). The major proportion corresponds to FDX2 bound in the distal conformation (65%) and only 14% of the particles show FDX2 in the proximal conformation. The remaining 20% of the dataset corresponds to the FXN-bound core ISC complex (abbreviated (NIAUX)$_2$), in which FXN is bound in a position virtually identical to that observed in previously published cryo-EM structures[18,19].

### FDX2 interacts with a conserved arginine patch of NFS1

In both proximal and distal conformations FDX2 forms salt bridges with a conserved arginine patch (Arg272, Arg275, Arg277) and with Arg145, Arg273 and Arg289 of NFS1' (NFS1 sequence alignment in Supplementary Fig. 6). The contacts are made to acidic residues of FDX2 helix F (Glu134, Asp137, Asp138, Asp141) and Glu148$^{FDX2}$, which interacts with Arg145$^{NFS1'}$ (Fig. 2c,d). In the distal conformation, fewer salt bridges are formed with NFS1' than in the proximal conformation (Fig. 2c) and the overall resolution of FDX2 is lower than for the rest of the complex (Supplementary Fig. 1e), indicating a more flexible binding mode in the distal position than in the proximal conformation, where FDX2 appears well resolved (Supplementary Fig. 1d).

The acidic patch of FDX2 helix F is highly conserved (FDX2 sequence alignment in Supplementary Fig. 7) and was previously suggested as the binding interface with NFS1 based on NMR experiments using the bacterial homologues[29,30]. This region additionally acts as the primary binding interface with FDXR, as determined for FDX1[34]. The NFS1 arginine patch has also been previously proposed to serve as a nuclear localization signal[35], although we note that the motif is also conserved in bacterial sequences (Supplementary Fig. 6).

### Proximal binding of FDX2 positions its [2Fe-2S] cluster for efficient electron transfer

In the distal conformation, the [2Fe-2S] cluster of FDX2 is too far for efficient electron transfer to the ISCU2 FeS cluster assembly site

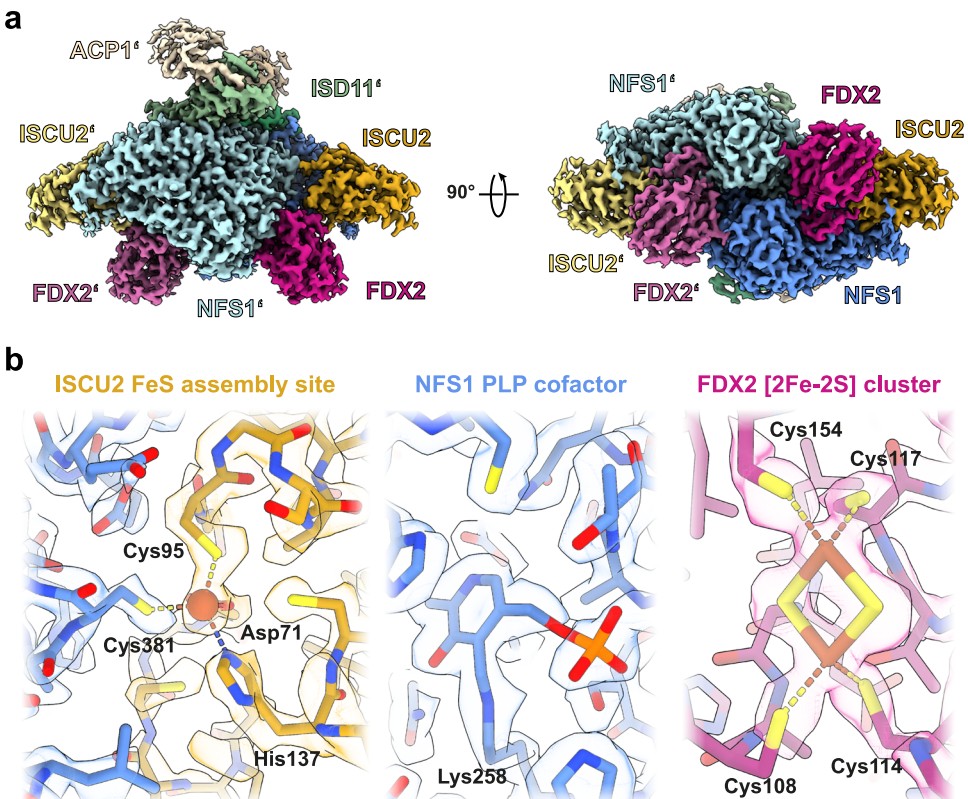

**Fig. 1 | Overall architecture and cryo-EM density details of the (NIAUF)$_2$ complex. a** The 2 Å consensus cryo-EM map (C2 symmetry applied) is segmented and colored by subunit. Except where otherwise indicated, subunit coloring is consistent throughout the manuscript. FDX2 binds in a cavity at the NFS1-ISCU2 interface. **b** Atomic model and density details in the FDX2-bound proximal structure (density-modified). (**Left**) ISCU2 FeS assembly site with the Fe$^{2+}$ ion (red sphere) coordinated by Cys381[NFS1], Cys95[ISCU2], Asp71[ISCU2] and His137[ISCU2]; (**middle**) NFS1 PLP cofactor covalently linked to Lys258[NFS1]; (**right**) FDX2 [2Fe-2S] cluster coordinated by Cys108, Cys114, Cys117 and Cys154.

(23 Å)[36], whereas the movement to the proximal binding position brings the [2Fe-2S] cluster to a distance of 14 Å that would allow rapid electron transfer for reduction of the Cys138[ISCU2] persulfide (Fig. 2b). In the distal binding position, the C-terminal 15 residues of FDX2 are unresolved, indicating that these residues are highly flexible when FDX2 is bound in this conformation. Disordered C termini are common among ferredoxins, and this region had to be truncated in human FDX2 to facilitate its crystallization (PDB 2Y5C)[11]. When bound in the proximal position, the C terminus of FDX2 is ordered and well resolved, interacting extensively with the NFS1 subunit (Fig. 3a,b). Contact is mediated by hydrogen bonds (H-bonds) between Asn175[FDX2] and Cys-loop Ser385[NFS1] (Fig. 3c), a salt bridge between Asp179[FDX2] and Arg393[NFS1] (Fig. 3d) and hydrophobic interactions of Phe176[FDX2] and Val178[FDX2] with Leu160[NFS1]. The interacting residues are conserved among higher eukaryotes (sequence alignments in Supplementary Figs. 6 and 7), which suggests the functional importance of these interactions (see below). In contrast, other conserved C-terminal residues of FDX2 do not seem to make specific contacts with NFS1 or ISCU2 subunits. Therefore, the function of this well-conserved region may not solely be in the recognition of the core ISC complex.

The loop region of ISCU2 (residues 66-71 including Cys69[ISCU2]) rearranges in correlation with the two FDX2 conformations. When FDX2 is bound in the distal conformation, Cys69[ISCU2] is shifted away from the ISCU2 FeS cluster assembly site (Fig. 4a, Supplementary Movie 1), whereas it is facing towards the iron, but too far for coordination (3.9 Å) when FDX2 is bound in the proximal conformation (Fig. 4b). The functional implication of this rearrangement remains to be clarified.

## Ferredoxin and frataxin bind to overlapping binding sites during turnover

The conserved arginine residues of NFS1' (Arg272, Arg275, Arg277, Arg289), which electrostatically interact with FDX2, also serve as the binding site for FXN (primarily via its residues Glu108, Glu111, Glu121 and Asp124)[18,37]. It was previously suggested that ferredoxin and frataxin compete for binding in homologous bacterial and yeast complexes[29–31]. In the bacterial system, Kim et al.[30] found that ferredoxin, IscU and CyaY (FXN homolog) compete for an overlapping binding site on IscS (NFS1 homolog). This does not appear to be the case for the human core ISC complex, as our cryo-EM structures show ISCU2 being tightly bound in the presence of FDX2 or FXN.

In our FXN-bound (Fe-NIAUX)$_2$ structure, the Cys-loop of NFS1 is predominantly facing the ISCU2 assembly site with Cys381[NFS1] coordinating the iron (Fig. 4c), but additional weak density in this region may originate from a low-occupancy alternate conformation of the Cys-loop, consistent with its high flexibility. FXN interacts with His137[ISCU2] and Cys69[ISCU2], which was previously observed for the Zn$^{2+}$ and Fe$^{2+}$-bound (NIAUX)$_2$ structures[18,19] and likely modulates the ISCU2 assembly site to facilitate persulfide transfer to Cys138[ISCU2] (ref. 19). In contrast, the structures of FDX2 bound in both conformations show His137[ISCU2] facing the ISCU2 FeS cluster assembly site in coordinating distance to Fe$^{2+}$ (Fig. 4a,b). This orientation of His137[ISCU2] is more similar to the structures of non-FXN-bound (NIAU)$_2$ complexes[13,19]. We note that the lower local resolution at the Fe-binding site, likely caused by its structural malleability, may mean that additional (solvent) ligands of Fe are not resolved.

All our FDX2-bound structures show the NFS1 Cys-loop in the 'outward' conformation where Cys381[NFS1] coordinates the ISCU2-

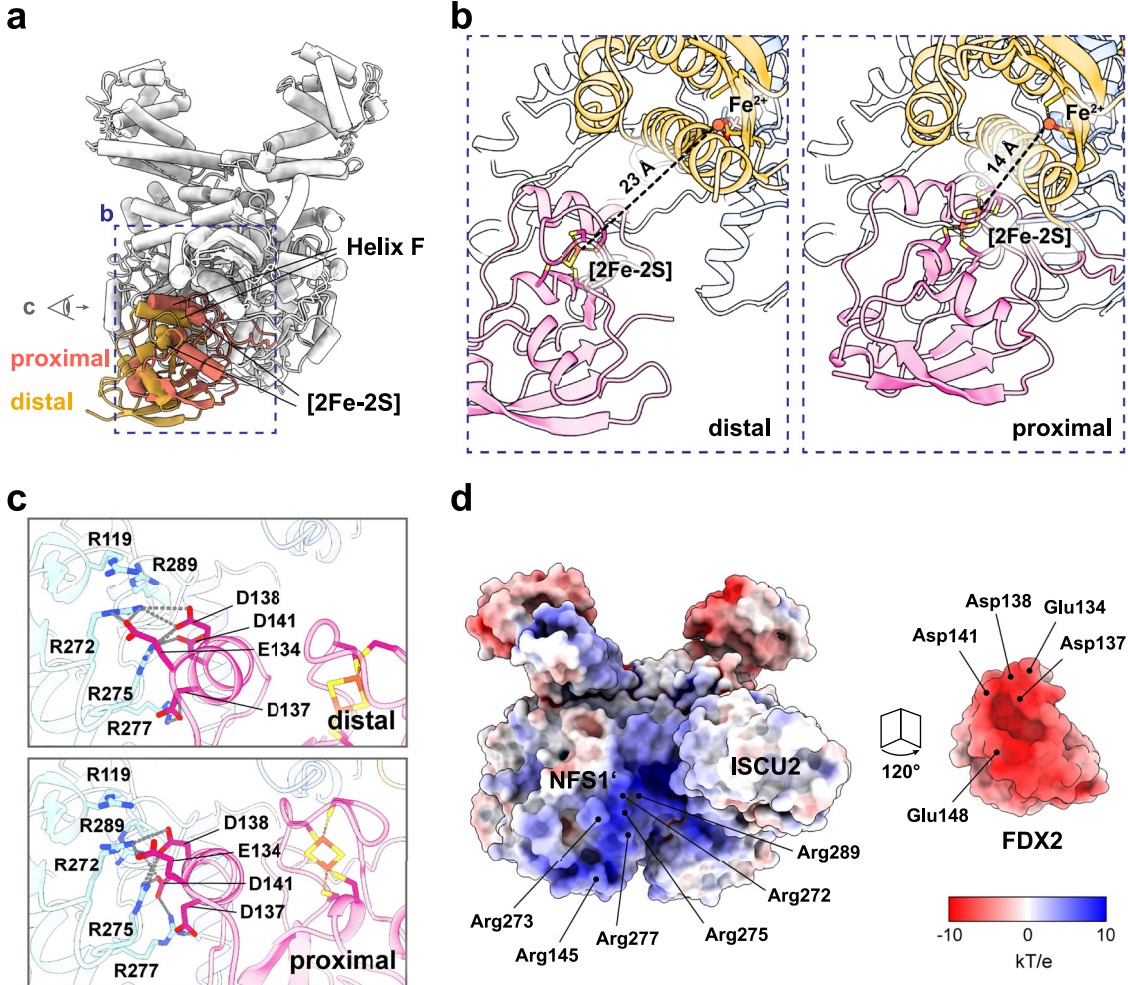

**Fig. 2 | FDX2 binds the (NIAU)$_2$ complex mainly via an arginine patch on NFS1'.**
**a** Overlay of the FDX2 distal (gold) and FDX2 proximal (orange) conformations. The conformational changes are visualized in Supplementary Movie 1. Residues on helix F form salt bridges with NFS1'. The dashed blue box and the eye symbol represent the views shown in b and c, respectively. **b** The distance between the FDX2 [2Fe-2S] cluster and the ISCU2 assembly site iron is 23 Å in the distal conformation. Binding of FDX2 in the proximal conformation decreases the distance to 14 Å. **c** Salt bridge interaction between FDX2 helix F and NFS1' in the distal and proximal conformations. Distances <4 Å are indicated by dashed grey lines. **d** Electrostatic potential representation of the (NIAU)$_2$ complex (**left**) and FDX2 (**right**). Residues involved in salt-bridge interactions between the patches are labelled.

bound Fe$^{2+}$ together with Asp71$^{ISCU2}$, His137$^{ISCU2}$ and (proximal conformation only) Cys95$^{ISCU2}$ (Fig. 4a, b). When FDX2 is bound in the proximal position, the H-bonds between Asn175$^{FDX2}$ and Ser385$^{NFS1}$ could provide additional stability for the 'outward' Cys-loop conformation. An outward-facing conformation of the Cys-loop was also observed in the X-ray structure of Zn$^{2+}$-bound (NIAU)$_2$ (ref. 13), indicating that this may be a resting conformation when neither FXN nor FDX2 are bound to the (NIAU)$_2$ complex.

### Mutational and functional testing of the conserved FDX2 C terminus

The second aim of the study was to biochemically elucidate the significance of the close contacts between FDX2 and other ISC factors within the (Fe-NIAUF)$_2$ structures. In particular, we examined the roles of i) the FDX2 C terminus that binds to NFS1 in the proximal conformation of the (Fe-NIAUF)$_2$ complex, ii) the salt bridges between FDX2 and NFS1 engaged in both distal and proximal conformations, and iii) the FDX2 residues near to ISCU2 (Supplementary Fig. 8). First, we evaluated the relevance of the FDX2 C terminus for functionality of the core ISC complex, which we hypothesize may play an important

role in the transition from the distal to the proximal conformation. These residues were shown to have functional relevance[11], and differ from other mitochondrial [2Fe-2S] ferredoxin family members, such as mammalian FDX1 or fungal Yah1 proteins (Fig. 5a). We generated FDX2 truncation variants lacking 1, 5, 10, or 12 C-terminal residues (termed ΔC1, ΔC5, ΔC10, ΔC12; Fig. 5a), encompassing the majority of residues that are unstructured in the distal conformation. The variants were purified after expression in *E. coli* and contained a wild-type equivalent of [2Fe-2S] clusters as indicated by CD and UV-Vis spectroscopy (Supplementary Figs. 9 and 10). Their [2Fe-2S] cluster could be efficiently reduced by NADPH and FDXR (Supplementary Figs. 9 and 10), indicating no rate-limiting effect of these C-terminal truncations for FDXR reduction under the conditions of a CD spectroscopy-based [2Fe-2S] cluster synthesis assay[10,22].

This assay was employed to record the activity of the FDX2 variants in enzymatic reconstitution of a [2Fe-2S] cluster on ISCU2 by (NIA)$_2$ and FXN, recording cluster formation at 431 nm[10]. All truncated variants were able to generate wild-type amounts of [2Fe-2S] clusters in a FDX2-dependent fashion (Fig. 5b and Supplementary Fig. 11a,b). These findings clearly show that the C-terminal part of FDX2 is not

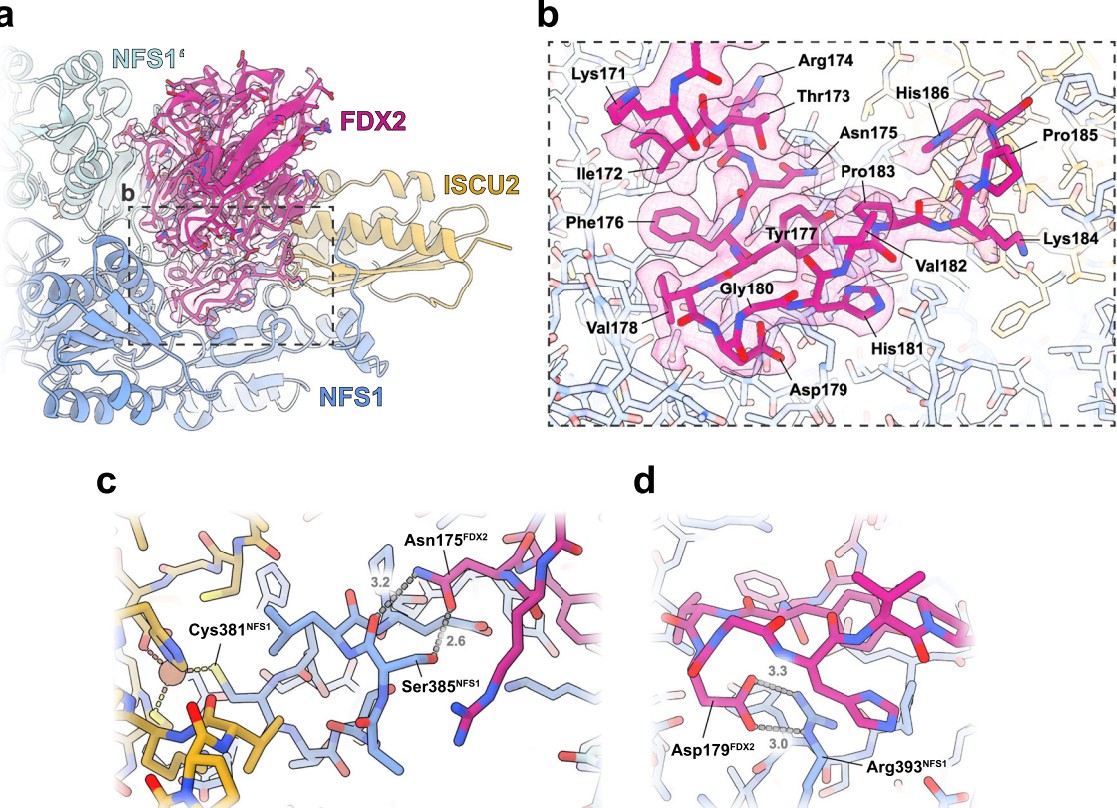

**Fig. 3 | Density and atomic model of the FDX2 C terminus. a** In the proximal conformation of the (NIAUF)$_2$ structure the FDX2 C terminus is resolved. The dashed grey box represents the view shown in b. **b** Atomic model of the FDX2 C terminus and corresponding map, prior to density modification. Side-chain density can be observed for nearly all residues, except for Lys184, Pro185 and His186. (Only FDX2 residues 171–186 are displayed for figure clarity). **c** H-bonds can form between Asn175$^{FDX2}$ of the FDX2 C terminus and Ser385$^{NFS1}$ of the NFS1 Cys-loop. **d** Salt-bridge interaction occurs between Asp179$^{FDX2}$ and Arg393$^{NFS1}$.

essential for in vitro [2Fe-2S] cluster assembly. This view is supported by the finding that fungal-type ferredoxins from *Saccharomyces cerevisiae* (ScYah1) or *C. thermophilum* (CtYah1) with C termini different from FDX2 can also support the in vitro [2Fe-2S] cluster synthesis, as can *T. hominis* Yah1, which contains an FDX2-like C terminus (Fig. 5a, red arrowheads; Fig. 5d; Supplementary Figs. 9 and 10).

Unexpectedly, the cluster assembly rates for ΔC1, ΔC5, and ΔC10 variants were up to fourfold higher than that for wild-type FDX2 under these standard conditions, i.e. at 5 μM FDX2 (Fig. 5b,c). Upon further truncation (ΔC12 variant) the [2Fe-2S] cluster assembly rate was only 1.5-fold higher than that of wild-type FDX2. To better understand these differences between wild-type and variant FDX2 proteins, we varied their concentrations during the cluster synthesis assay. For wild-type FDX2, we found a bell-shaped behavior for [2Fe-2S] cluster synthesis with a maximum rate at around 1-2 μM FDX2, i.e. much lower than the standard concentration (5 μM) used above, where FDX2-dependent reduction is not rate-limiting (Fig. 5e). Above 2 μM FDX2, cluster synthesis rates decreased with increasing concentrations of FDX2, a result reminiscent of studies in yeast showing a competitive behavior between Yah1 and Yfh1 (see below; ref. 31). The three titration curves of the ΔC1, ΔC5, and ΔC10 variants were similar, with maximum synthesis rates over a broad concentration range of 2-5 μM, which was higher than that of wild-type FDX2 (Fig. 5e). Further increasing the variant concentrations to 10 μM decreased the synthesis rates similarly to those observed for wild-type FDX2, meaning the large rate differences to wild-type FDX2 remained (Fig. 5c,e,f and Supplementary Fig. 11c). Conspicuously, the kinetic advantage of the C-terminal variants over wild-type FDX2 was lost at concentrations lower than ca. 1.5 μM (Fig. 5e,g and Supplementary Fig. 11e). The similar effects elicited by these three variants suggested that mainly the C-terminal His186$^{FDX2}$

was responsible for decreasing the rate of cluster synthesis at high FDX2 concentrations. Residue His186$^{FDX2}$ is resolved only in the proximal conformation of our (Fe-NIAUF)$_2$ structure, yet does not interact with other residues in this complex. Hence, this structure does not provide any molecular clue for how this residue might affect the overall synthesis reaction.

The ΔC12 titration curve was markedly different from both WT FDX2 and the other C-terminal variants, with the comparatively low rate of [2Fe-2S] cluster synthesis increasing up to a FDX2 concentration of 10 μM (Fig. 5e,f), indicating that either N175$^{FDX2}$ (hydrogen bonding to Ser385$^{NFS1}$; Fig. 3c; see also below) and/or Phe176$^{FDX2}$ are important though not essential determinants for cluster synthesis. Since we did not find differences between the ΔC5 and ΔC10 variants, we conclude that the influence of the salt bridge formed between Asp179$^{FDX2}$ and Arg393$^{NFS1}$ on the synthesis rate is minor (cf. Figures 3d and 5a).

The assay results for the ΔC12 mutant and the interaction of Asn175$^{FDX2}$ with Ser385$^{NFS1}$ via hydrogen bonds in the proximal conformation suggested an important role for this residue. To directly examine this idea, we tested the site-specific mutant protein FDX2-N175A (Asn exchanged for Ala) after purification from *E. coli* and verification of cluster content and FDXR reducibility by CD and UV-Vis spectroscopy (Supplementary Figs. 9 and 10). Strikingly, FDX2-N175A efficiently supported [2Fe-2S] cluster assembly only at higher concentrations and behaved similar to the ΔC12 variant upon titration (Fig. 5h). This result demonstrates that residue Asn175$^{FDX2}$ is important but not essential for normal function of FDX2, because its exchange can be compensated by increasing FDX2-N175A concentrations. Based on these findings and our cryo-EM structures, Asn175$^{FDX2}$ may be a decisive determinant for the transition from the distal to the proximal conformation.

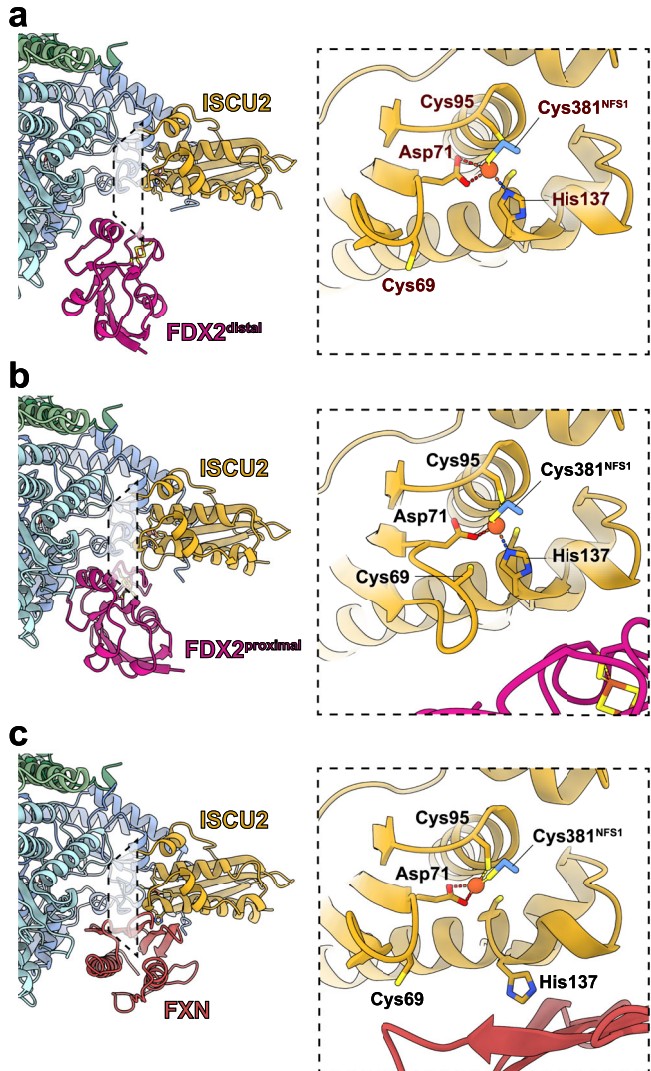

**Fig. 4 | Local rearrangements near the FeS cluster assembly site in the presence of FDX2 or FXN. a** When FDX2 is bound in the distal conformation, the iron is coordinated by Asp71[ISCU2], His137[ISCU2] and Cys381[NFS1]. Cys95[ISCU2] faces the iron, but is too far for coordination (3.5 Å). Cys69[ISCU2] faces away from the ISCU2 FeS assembly site, similar to the FXN-bound structure. **b** When FDX2 is bound in the proximal conformation, the iron is coordinated by Asp71[ISCU2], Cys95[ISCU2], His137[ISCU2] and Cys381[NFS1]. Cys69[ISCU2] faces towards the iron, but is too far for coordination (3.9 Å). **c** In the FXN-bound structure, the iron in the ISCU2 FeS assembly site is coordinated by Asp71[ISCU2], Cys95[ISCU2] and Cys381[NFS1]. We note that the local resolution does not allow us to exclude the possibility of solvent molecules as additional ligands to the Fe.

We reasoned that the bell-shaped behavior of the cluster synthesis rates during FDX2 concentration variation (Fig. 5e) may be caused by the competition of FDX2 and FXN for the same binding site, as suggested by cryo-EM structures (this work and refs. 18,19) and by recent biochemical studies in yeast[31]. Hence, optimal relative concentrations of both ISC proteins may be needed for maximum synthesis rates, and altering the FDX2 C terminus may affect this competition. We tested if FXN titrations may also show a competitive behavior in the cluster synthesis assay. A maximum synthesis rate was reached above 1 μM FXN (at 2.5 μM FDX2). The rate hardly changed up to 10 μM FXN, yet dropped substantially upon further increasing the FXN concentration to 20 μM (Fig. 5i). This result shows that also FXN exhibits a bell-shaped concentration dependence, yet the much higher concentration needed to decrease the cluster synthesis rates indicates that FXN is a comparatively weaker competitor under turnover.

## Functionally crucial electrostatic interactions between NFS1 and FDX2

The role of the salt bridges between helix F of FDX2 and the Arg-rich region of NFS1' was investigated (cf. Figure 2c,d and Supplementary Fig. 8). We exchanged five Arg residues of NFS1 (Arg271, Arg272, Arg273, Arg275, and Arg277) to Ala (NFS1-RA variant) or Asp137-Asp138 of FDX2 to Ala (FDX2-DA variant) and purified these proteins from *E. coli* (sequence alignments in Supplementary Figs. 6 and 7). NFS1-RA did not support any significant [2Fe-2S] cluster synthesis, consistent with this region's double function in binding both FXN and FDX2 via Arg residues[31] (Fig. 6a). The important FXN-connected role of the Arg patch became evident from a FDX2-independent assay, the FXN- and ISCU2-stimulated (non-physiological) production of free sulfide by (NIA)₂ in the presence of the reductant dithiothreitol (DTT)[15,22]. For wild-type NFS1, but not for NFS1-RA, the reaction rate was stimulated by the addition of FXN and ISCU2 (Fig. 6b). NFS1-RA was still able to produce a basal amount of sulfide, indicating that the protein is still active in persulfidation.

Since the double function of the NFS1 Arg residues in FDX2 and FXN binding did not allow us to clearly dissect the importance of FDX2 by replacement of these residues, we investigated the FDX2-DA variant carrying exchanges of NFS1-binding residues. The protein had a wild-type [2Fe-2S] cluster content (Supplementary Fig. 12a), yet could not be reduced by NADPH-FDXR, likely because Asp residues are involved in binding to FDXR, as suggested by an X-ray structure of FDX1-FDXR (PDB 1E6E[34]; Supplementary Fig. 12a,b). Accordingly, even high concentrations of FDX2-DA in the standard FDXR-dependent assay did not give rise to any [2Fe-2S] cluster formation (Fig. 6c). We therefore used the chemical reductant sodium dithionite (DT) which could reduce both wild-type FDX2 and the FDX2-DA variant with similar efficiency and time dependence (Supplementary Fig. 12c,d). When the DT-reduced proteins were compared in the CD-based cluster synthesis assay, both were able to generate [2Fe-2S] clusters on ISCU2, yet the FDX2-DA-assisted synthesis rate was fourfold slower, indicating a decisive role of these Asp residues in the NFS1-FDX2 functional interaction (Fig. 6d,e). We note, however, that the DT-supported reaction with wild-type FDX2 differed from the standard FDXR-assisted reaction in various aspects (Fig. 6d). The former occurred at higher rate, yet with an overall lower synthesis efficiency. Further, the CD spectra of the ISCU2-bound [2Fe-2S] product differed characteristically for both FDX2 and FDX2-DA proteins from that of wild-type FDX2 generated in the standard FDXR-supported reaction (Supplementary Fig. 12e). This difference was not due to a potential DT-induced alteration of cluster coordination on ISCU2, because the ISCU2-bound [2Fe-2S] spectrum obtained by the standard enzymatic reaction (i.e. by FDX2-FDXR) did not change upon subsequent DT treatment, apart from mild bleaching (Supplementary Fig. 12f). The differences between the enzymatic and the DT-supported cluster formation call for caution in the use of this reductant. Nevertheless, our results suggest the importance of the salt bridges between NFS1 and FDX2 for [2Fe-2S] cluster synthesis.

## The contact area of FDX2 and ISCU2 is not important during [2Fe-2S] cluster assembly

Finally, we exchanged FDX2 residues located in the vicinity of ISCU2 (residues 109EASL112; Fig. 5a). Even though these residues do not directly touch ISCU2 in the proximal conformation of the (Fe-NIAUF)₂ complex and specific interactions are not obvious, a slight movement of FDX2 towards ISCU2 may lead to contacts (Supplementary Fig. 8a). Residues EASL are part of the 'cluster-binding loop' of FDX2, and hence their exchange could possibly affect cofactor binding. To minimize the possible effects of amino acid exchanges on cluster binding, we replaced them with residues GGSC present in fungal Yah1 proteins (Fig. 5a). The resulting variant FDX2-CL ('CL' for 'cluster-binding loop') exhibited a cluster content similar to WT and was reducible by NADPH-FDXR (Supplementary Fig. 9 and 10). Moreover, FDX2-CL was able to

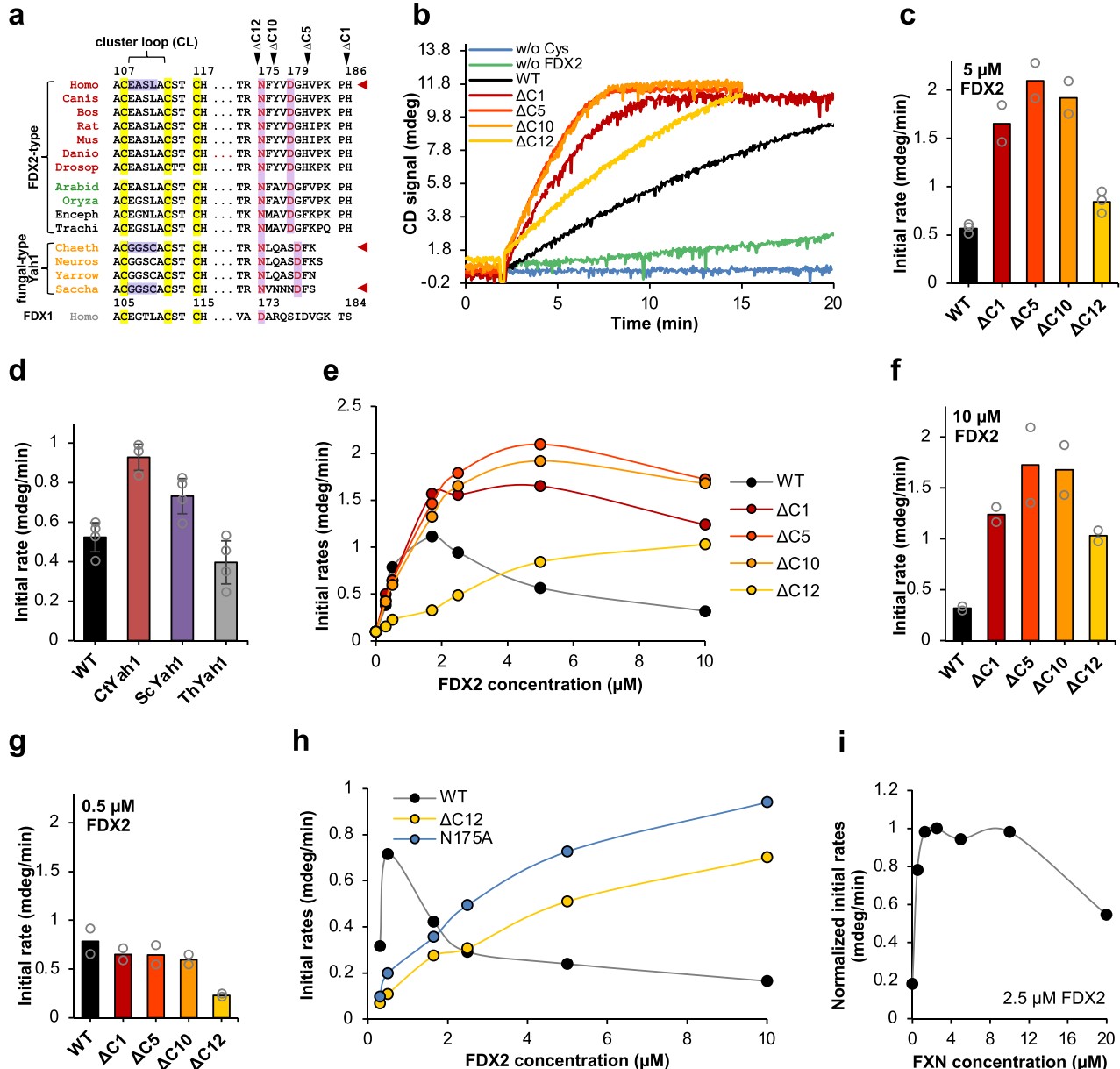

**Fig. 5 | Effects of FDX2 C-terminal mutations on [2Fe-2S] cluster synthesis.**
**a** Multi-sequence alignment of different ferredoxin types showing the rationale for amino acid exchanges (light blue) in the cluster-binding loop (CL) or truncations within the C terminus of human FDX2. ΔC indicates the number of deleted C-terminal residues. [2Fe-2S] cluster-binding Cys residues are highlighted in yellow. Conserved residues $Asn175^{FDX2}$ and $Asp179^{FDX2}$ interact with NFS1. **b** Time-resolved enzymatic in vitro [2Fe-2S] cluster synthesis with 3 μM (NIA), 30 μM ISCU2, 5 μM FXN, and 5 μM of wild-type (WT) or indicated variant FDX2 (see part a) was measured anaerobically by CD spectroscopy. Control reactions lacked either FDX2 or Cys (w/o). **c** Quantitation of the initial rates from part b (for WT, ΔC12: mean of $n = 3$; for ΔC1, ΔC5, ΔC10: mean of $n = 2$; see also Supplementary Fig. 11b). **d** Initial rates of [2Fe-2S] cluster synthesis as in **b** with 5 μM each of FDX2 or the indicated fungal Yah1 ferredoxins from Ct, *C. thermophilum*; Sc, *S. cerevisiae*; or Th, *T. hominis*. Values are the mean ± SD (for WT, ScYah1, ThYah1: n = 4; for CtYah1: $n = 3$; technical replicates). **e** Initial rates of standard [2Fe-2S] cluster synthesis (see b) with varying concentrations of WT or variant FDX2 as indicated. Data points (mean from $n ≥ 2$ individual measurements) were connected by Akima interpolation for clarity. **f, g** Comparison of the rates from (**e**) at 10 μM (**f**) and 0.5 μM (**g**) FDX2 (mean of $n = 2$; see also Supplementary Fig. 11c-e). **h** Initial rates of [2Fe-2S] cluster synthesis (see **e**) with varying concentrations of WT FDX2, ΔC12, or FDX2-N175A (for WT 5 μM: mean of $n = 4$; for N175A: mean of $n = 2$; for others: $n = 1$). Data points were connected by Akima interpolation. **i** Initial rates of cluster synthesis in the presence of varying concentrations of FXN (normalized to 2.5 μM FXN; $n = 1–3$). Data points were connected by Akima interpolation. Source data are provided as a Source Data file.

support [2Fe-2S] cluster assembly similar to wild-type FDX2 with a maximum synthesis rate at 1.5 μM, yet the rate was twofold slower than wild-type (Fig. 6c). Therefore, these residues are not essential for a potential interaction of FDX2 with ISCU2.

Collectively, our mutational and biochemical investigations of the contact areas of FDX2 within the core ISC complex revealed functionally important interactions. Salt bridges between FDX2 and NFS1 are engaged in both distal and proximal conformations of the (Fe-

$NIAUF)_2$ complex, and hydrogen bonding of C-terminal $N175^{FDX2}$ and $Ser385^{NFS1}$ may contribute to the transition from the distal to the proximal conformation. The C terminus is not essential for this interaction despite its high conservation. Our structures show minimal contact between FDX2 and ISCU2, and exchanges of potential contact site residues suggest their interaction plays a minor if any role for in vitro [2Fe-2S] cluster synthesis, despite the fact that these proteins have been shown to tightly interact both in vitro and in vivo, in

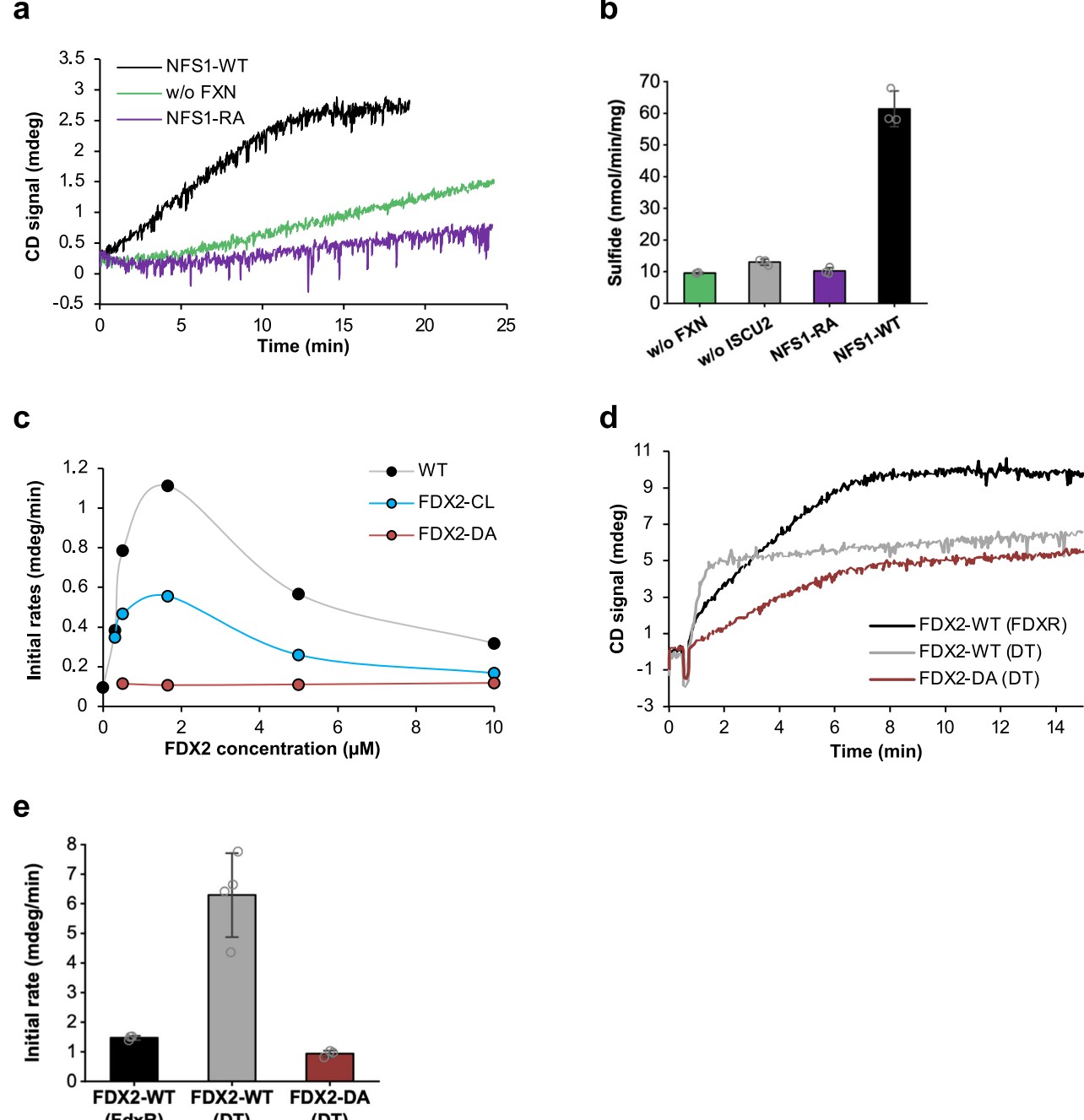

**Fig. 6 | Functional impact of mutations in helix F or the cluster-binding loop of FDX2. a** Time-resolved enzymatic in vitro [2Fe-2S] cluster synthesis was measured by CD spectroscopy as in Fig. 5b using wild-type (WT) NFS1 or the NFS1-RA variant. A control reaction lacking FXN was included (w/o FXN). **b** Determination of DTT-mediated sulfide (S²⁻) production by WT NFS1 or the NFS1-RA variant in the presence or absence of ISCU2 and FXN as indicated (mean ± SD of n = 3; technical replicates). **c** Initial rates of [2Fe-2S] cluster synthesis with varying concentrations of WT FDX2 (*n* ≥ 2) or the FDX2-CL (*n* = 2) and FDX2-DA (*n* = 1) variants. Data points were connected by Akima interpolation. **d** Time-resolved [2Fe-2S] cluster synthesis with dithionite (DT)-reduced WT FDX2 or the FDX2-DA variant. A standard cluster synthesis reaction with NADPH-FDXR-reduced WT FDX2 served as a reference. **e** Quantitation of the initial rates from part d; values are mean ± SD (for FDX2-WT (DT): *n* = 4; FDX2-WT (FDXR), FDX2-DA (DT): *n* = 3; technical replicates). Source data are provided as a Source Data file.

particular in the reduced form of FDX2 or ScYah1[10]. Such an interaction may involve other parts of the proteins[10] and may be connected to a step after [2Fe-2S] cluster assembly.

## Discussion

Our cryo-EM structures reveal the binding modes of FDX2 to the human core ISC complex at near-atomic resolution. FDX2 interacts electrostatically with a series of arginine residues on NFS1′, overlapping with the binding position of FXN. This finding is consistent

with previous studies that observed competitive binding of the bacterial FDX2 and FXN orthologs[29,30]. We did not observe any evidence for a dodecameric core ISC complex, in which FDX2 and FXN would bind simultaneously, as suggested by a previous model[13]. This model was established on the basis of SAXS envelopes obtained from various reconstitutions using components of the *C. thermophilum* core ISC complex[13] taking into consideration interacting residues between *C. thermophilum* apo-Isu1 (ISCU2 homolog) and chemically reduced yeast Yah1 (FDX2 homolog) that were identified by NMR spectroscopy[10]. In

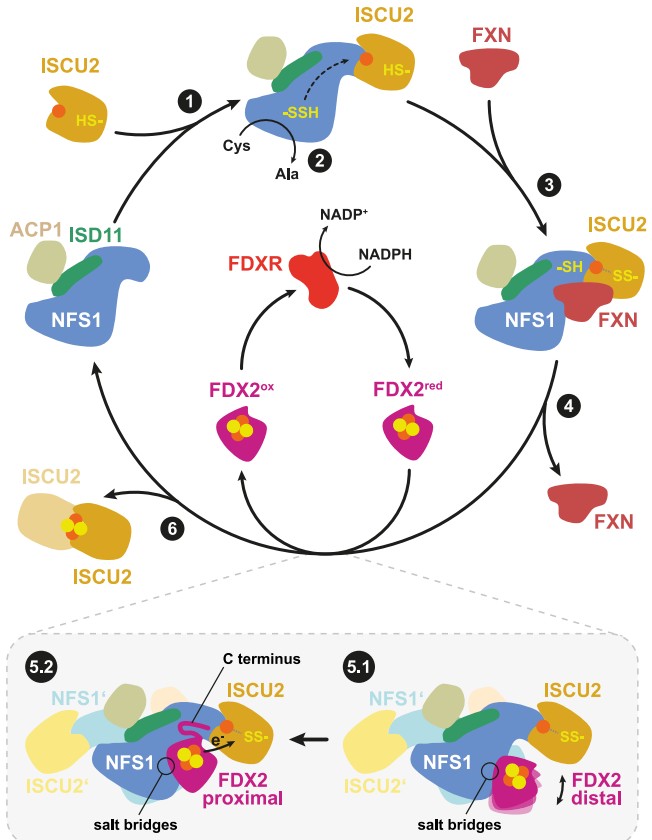

**Fig. 7 | Scheme of the major steps of de novo [2Fe-2S] cluster biogenesis by the core ISC complex.** For simplicity the second dimer half is shown only for step 5. (**1**) The (NIA)$_2$ subcomplex binds to the FeS scaffold protein ISCU2 to form the (NIAU)$_2$ complex. (**2**) Desulfuration of free cysteine (Cys) at the PLP cofactor yields a protein-bound persulfide, which is transferred to ISCU2 via Cys381[NFS1] of the flexible Cys-loop. (**3**) Transient interaction with the allosteric activator FXN modulates the ISCU2 assembly site to facilitate sulfur transfer from Cys381[NFS1] to Cys138[ISCU2]. (**4**) FXN leaves the complex to allow FDX2-binding. (**5.1**) Reduced FDX2 binds loosely to the NFS1 arginine patch in the distal conformation. (**5.2**) Rigidification of the flexible FDX2 C terminus allows FDX2 to shift into the more rigidly bound proximal conformation where the [2Fe-2S] cluster is close enough for transfer of an electron (e⁻) to the ISCU2 FeS cluster assembly site. Oxidized FDX2 is reduced by FDXR and NADPH. (**6**) The [1Fe-1S] intermediate is converted into a mature [2Fe-2S] cluster by dimerization of two ISCU2 subunits.

this model, FDX2 is suggested to interact with ISCU2 in a location that is occupied by the NFS1 C terminus in the cryo-EM structures presented here and by Fox et al.[18]. In our present study, the tight and specific interaction between NFS1 and FDX2 via salt bridges and the rigidification of the otherwise flexible C-terminal residues supports the idea that this is the physiological binding site of FDX2 within the core ISC complex during FeS cluster biogenesis (Fig. 7).

An established CD-based assay of [2Fe-2S] cluster synthesis[10] supports the structural findings of an overlap of the binding sites of FDX2 and FXN within the core ISC complex, implying a sequential and competitive action of FXN and FDX2 during the cluster synthesis process (Fig. 7). Biochemically, this feature appears to be reflected in the bell-shaped concentration dependence for both proteins in our cluster synthesis assay. This scenario fits well to in vitro findings for the competitive behavior of yeast proteins Yah1 and Yfh1 in (NIAU)$_2$ binding studies[31]. The lower optimal concentration of human FDX2 compared to FXN indicates that FXN is a weaker competitor than FDX2. Our findings are consistent with a quantitative proteomics study that found ~5-fold higher abundance of FXN than FDX2 in human mitochondria[38]. The fact that FDX2 and FXN compete for an

overlapping binding site may have implications for understanding Friedreich's ataxia, which is most commonly due to lowered abundance of FXN, rather than differences in its amino acid sequence[25]. Future biochemical studies need to address which thermodynamic and kinetic parameters underlie the binding and dissociation of FDX2 and FXN, while in vivo studies will be required to assess the physiological importance of this competitive effect.

Our structures show that FDX2 binds in two conformations, termed 'proximal' and 'distal' according to the distance of the electron-donating [2Fe-2S] cluster of FDX2 to the ISCU2 FeS cluster assembly site, where Fe²⁺ and the persulfide bind (Fig. 7). During the transition from the distal to the proximal conformation, the FDX2 [2Fe-2S] cluster moves by a distance of around 8 Å, thereby diminishing the distance to the ISCU2 FeS cluster assembly site from 23 to 14 Å, a distance that would allow efficient electron transfer. In the proximal conformation the conserved C-terminal residues of FDX2 become ordered and interact with NFS1. Transitions between the two conformations involve a rotational motion around FDX2 helix F (Supplementary Movie 1), which contains negatively charged residues that form electrostatic interactions with several arginine residues of NFS1. The movement is achieved by a rearrangement and tightening of the salt bridges between NFS1' and FDX2. A possible explanation for the existence of two binding modes for FDX2 could be that the distal conformation is kinetically favored and represents a first stage of binding, which then enables the intrinsically disordered C terminus of FDX2 to rigidify and transiently bind to NFS1.

We biochemically investigated the significance of the structural contacts of FDX2 within the core ISC complex by employing a combination of mutational and functional approaches. In addition to the interactions of the C-terminal region and the salt bridges of FDX2 with NFS1 as observed in the structural context, we tested a potential interaction site between FDX2 and ISCU2, although not even in the proximal cryo-EM structure a specific direct contact is observed. FDX2 variants carrying alterations in all three contact areas retained the ability to bind and reduce the endogenous [2Fe-2S] cluster. We then functionally evaluated these variants biochemically using an established CD-based [2Fe-2S] cluster biosynthesis assay[10]. The mild effect observed upon exchange of the potential interaction area of FDX2 with ISCU2 and the lack of obvious specific contacts between the two proteins within the proximal (Fe-NIAUF)$_2$ structure make it unlikely that this area is of functional importance for FDX2 binding.

The functional role of the conserved FDX2 C terminus was explored by creating a number of truncation variants including one that lacked this segment. Strong conservation of the C-terminal residues of 'type II ferredoxin' (FDX2) is a key difference that distinguishes FDX2 from type-I adrenodoxin (FDX1) and the fungal-type eukaryotic [2Fe-2S] ferredoxins[11,32]. The central role of FDX2 in FeS cluster biosynthesis[8] and its specificity for this process was shown to be partially related to the C-terminal 12 residues[11]. However, in our (Fe-NIAUF)$_2$ structure only two of the conserved C-terminal FDX2 residues (Asn175 and Asp179) displayed specific contacts to NFS1. Therefore, the strong conservation of the mammalian FDX2 C terminus may be needed for other physiological tasks of FDX2 (cf. ref. [11]), for instance for the fusion of two [2Fe-2S] to a [4Fe-4S] cluster by the ISCA1-ISCA2-IBA57 complex[23]. Addressing this interesting question will require in vivo studies in human cell culture.

Functional analysis of the C-terminal truncation variants demonstrated that this region is important but not essential for [2Fe-2S] cluster synthesis. This notion was independently confirmed by investigation of fungal FDX2 counterparts with different C termini in our human [2Fe-2S] cluster synthesis assay. These heterologous ferredoxin relatives were functional in our assay system with otherwise human ISC proteins. Comparison of the various C-terminal truncation variants identified two residues that are either critical for high synthesis rates (Asn175[FDX2]) or that negatively interfere with the rate of [2Fe-2S] cluster

synthesis at high concentrations of FDX2 (His186$^{FDX2}$). Another conserved residue Asp179$^{FDX2}$, despite forming a salt bridge to Arg393$^{NFS1}$ in the cryo-EM structure, exhibits a minor if any role in [2Fe-2S] cluster synthesis. Asn175$^{FDX2}$ was found to be crucial for high rates of [2Fe-2S] cluster synthesis, a finding which was independently supported by analyzing a site-specific variant. This effect can readily be explained by its hydrogen bonding with Ser385$^{NFS1}$, as seen in our (Fe-NIAUF)$_2$ structures. This specific interaction may facilitate the movement of FDX2 from the distal to the proximal conformation of FDX2 in the ISC complex, and/or may stabilize the outward-facing conformation of the NFS1 Cys-loop containing Ser385$^{NFS1}$. Understanding the mechanistic role of Asn175$^{FDX2}$ will require future in vivo and in vitro experiments. Removal of His186$^{FDX2}$ increases the rate of [2Fe-2S] cluster synthesis, but the interpretation of this effect remains enigmatic, the more so as this effect was observed only at higher (i.e. non-rate-limiting) FDX2 concentrations. His186$^{FDX2}$ is resolved only in the proximal conformation of the (Fe-NIAUF)$_2$ complex, yet does not entertain any specific interactions. It will therefore be important to explore the in vivo consequences of deleting this conserved residue.

For the salt bridges between FDX2 and NFS1, our data clearly indicate that this electrostatic interaction is crucial for biosynthetic function, because exchanges of charged residues in either FDX2 or NFS1 strongly impaired cluster synthesis. Even though the synthesis assay cannot resolve the relative importance of the binding modes in distal and proximal conformations, it seems likely to us that the two binding modes are used in a sequential fashion. Notably, the positively charged region of NFS1 that interacts with FDX2 is also important for FXN binding, as supported by a sulfide production assay which is independent of FDX2. In this assay, the NFS1-RA variant showed a basal (i.e. FXN-independent) rate of sulfide production similar to wild-type, indicating that impaired FXN binding rather than NFS1 dysfunctionality was responsible for the low sulfide production. Collectively, our results clearly demonstrate the crucial role of the electrostatic interaction between FDX2 and NFS1.

The distinct conformational changes in the ISCU2 FeS cluster assembly site upon FXN and FDX2 binding may be crucial to advance the partial reactions underlying [2Fe-2S] cluster biosynthesis. These steps include FXN binding facilitating persulfide transfer from Cys381$^{NFS1}$ to Cys138$^{ISCU2}$ as well as FDX2 binding allowing reductive conversion of the persulfide to sulfide and eventual [2Fe-2S] cluster formation by dimerization of two ([1Fe-1S]-NIAU)$_2$ entities[22]. Overall, our study has advanced the knowledge of [2Fe-2S] cluster synthesis at the mitochondrial core ISC complex and offers many new avenues for future exploration of this essential process of life.

## Methods

### Protein production and purification

The sequences of all proteins used in this study were derived from *H. sapiens* except for the Yah1 proteins from *C. thermophilum* (synonym for *Thermochaetoides thermophila* DSM 1495), *S. cerevisiae* and *Trachipleistophora hominis* (sequence information listed in Supplementary Table 1). They were recombinantly synthesized in *Escherichia coli* strain BL21 (DE3) and purified at 4 °C or on ice[22]. FDXR was produced and purified as described previously[23]. For all other proteins, cells were transformed with the appropriate plasmids (Supplementary Table 2) and grown at 37 °C for 8 h in terrific broth media. For FDX2 or (NIA)$_2$ expression, *E. coli* media were supplemented with 50 µM Fe(NH$_4$)$_2$(SO$_4$)$_2$ or 2 mM pyridoxine hydrochloride, respectively. Gene expression was induced at OD$_{600}$ ~ 0.8 by addition of either 1 mM isopropyl-β-D-thiogalactopyranoside (IPTG; Carl Roth GmbH) or 0.2 µM anhydrotetracycline (Sigma). Cells were grown overnight at 22 °C, harvested by centrifugation, and flash-frozen in liquid nitrogen. For purification of His-tagged proteins ((NIA)$_2$, ISCU2, ThYah1 and FXN), cells were thawed at room temperature and resuspended in

IMAC buffer (35 mM Tris-HCl pH 7.4, 300 mM NaCl, 5% (w/v) glycerol and 10 mM imidazole). FDX2 was purified by anion exchange chromatography (AEC), for which cells were resuspended in AEC buffer (35 mM Tris-HCl pH 7.5, 50 mM NaCl, 5% (w/v) glycerol). Protease inhibitor (cOmplete; Roche), lysozyme, and DNase I (and 5 mM PLP for (NIA)$_2$ purification) were added to the cell suspensions prior to cell lysis by sonication (SONOPULS mini20; BANDELIN electronic GmbH & Co. KG). Lysates were cleared by centrifugation at 40,000x g for 45 min. Supernatants of (NIA)$_2$, ISCU2, ThYah1 and FXN were subjected to Ni-NTA affinity chromatography (His-Trap 5 ml FF crude; GE Healthcare). The column was washed with 10 CV IMAC wash buffer, which contained 10 mM imidazole (for (NIA)$_2$) or 70 mM imidazole (for ISCU2 and FXN). Proteins were eluted with IMAC buffer containing 250 mM imidazole.

In the case of other ferredoxins and ferredoxin variants the supernatant was loaded onto an anion exchange column (Source 30Q) pre-equilibrated with AEC buffer and eluted with a linear gradient to high-salt buffer (35 mM Tris-HCl pH 7.5, 1 M NaCl, 5% (w/v) glycerol). Eluted proteins were concentrated using centrifugal concentrators (Amicon; 10-kDa molecular weight cutoff). For (NIA)$_2$ purification an additional AEC step was added after His purification to achieve higher purity for structural studies. FXN was treated with recombinant TEV protease (purified in-house) for removal of the N-terminal His-tag. (NIA)$_2$, FXN, and ISCU2 were transferred into an anaerobic chamber (Coy Laboratories) and incubated for 60 min with 10 mM DTPA, 10 mM TCEP, 25 mM KCN (for FXN and ISCU2) and 0.5 mM PLP (additionally for (NIA)$_2$) to remove metal ions and polysulfane sulfur before proteins were subjected to anaerobic size exclusion chromatography (SEC) (HiLoad 16/600 Superdex; GE Healthcare) on an Äkta Purifier system (GE Healthcare) using SEC buffer (50 mM Tris-HCl pH 8, 150 mM NaCl, 5% glycerol). Ferredoxins were purified further by aerobic SEC using the same SEC buffer, followed by an exchange to anaerobic buffer with PD10 columns in an anaerobic chamber.

For cryo-EM sample preparation ISCU2 was pre-incubated with FeCl$_2$ and ascorbate (both in 5x molar excess), FDXR was supplemented with MgCl$_2$ in 2x molar excess. Buffer was exchanged to 20 mM Tris-HCl pH 7.4, 100 mM NaCl for all proteins with PD10 columns (GE Healthcare) and protein concentrations were determined by the Bradford assay (Biorad). A correction factor for the resulting concentrations was determined for (NIA)$_2$, ISCU2, FXN, and FDX2 by quantitative amino acid analysis (Leibniz-Institut für Analytische Wissenschaften). Proteins were transferred into gas-tight glass vials and flash-frozen in liquid N$_2$.

### In vitro FeS cluster reconstitution on ISCU2

De novo synthesis of [2Fe-2S] clusters by the core ISC complex was performed as described[10,11] with minor modifications. If not indicated otherwise 3 µM NIA, 30 µM ISCU2, 5 µM FXN, 0.5 µM FDXR, and FDX2 in varying concentrations, with 0.8 mM sodium ascorbate, 0.5 mM FeCl$_2$, 0.1 mM MgCl$_2$ and 0.5 mM NADPH were mixed in SEC buffer, in an anaerobic chamber. The mixture was transferred to a sealed cuvette, with a magnetic stirrer and kept at 30 °C while the CD signal at 431 nm was recorded for about 15 min (Jasco J-815 Circular Dichroism Spectrophotometer). After approx. 2 min the reaction was started by addition of 0.5 mM L-Cys with a syringe, to a final reaction volume of 300 µL. Initial rates were determined in the linear range of each curve.

For the FDX2-DA variant experiments (Fig. 6c–e and Supplementary Fig. 12e,f) 0.8 mM Fe(NH$_4$)$_2$(SO$_4$)$_2$, 1 mM MgCl$_2$, 40 µM ISCU2 and 5 µM of each NIA, FXN and FDX2 were used and for chemical reduction 0.8 mM dithionite (DT) was added instead of FDXR and NADPH. Both mixtures were incubated 10 min at RT before starting the time course measurement, at 20 °C. After the enzymatic reconstitution with FDXR and recording of the post-reaction CD spectrum, 0.8 mM DT was

added to the assay mixture, which was incubated for 10 min at RT before another CD spectrum was recorded.

## Reduction of ferredoxin variants under anaerobic conditions

20 µM ferredoxin protein and 2 µM FDXR in SEC buffer supplied with 1 mM $MgCl_2$ were mixed anaerobically, transferred to a sealed cuvette and the UV/Vis and CD spectra were recorded. Under anaerobic conditions 0.2 mM NADPH was added, after mixing and 5 min incubation at RT the UV/Vis and CD spectra of the reduced states were recorded.

In the case of the FDX2-DA variant (Supplementary Fig. 12a-d) the enzymatic reduction was started by adding 0.8 µM FDXR in the CD spectrophotometer with a syringe and the CD signal was recorded at 436 nm. For the chemical reduction, 1.6 mM DT was added and the measurement was started after a 2 min incubation.

## Cysteine desulfurase activity determination by DTT-dependent sulfide release

Sulfide production by NFS1 and its RA variant (Fig. 6b) was measured spectrophotometrically essentially as described previously[22,39] measuring the dithiothreitol (DTT)-dependent sulfide generation. In brief, 0.6 µM WT or RA variant of $(NIA)_2$, 0.6 µM of ISCU2 and FXN were mixed in 25 mM Tricine pH 8.0, 1 mM DTT, and the reaction was started by adding 1 mM L-Cys. After 20 min at 30 °C, the reaction was quenched by 4 mM N,N-dimethyl-p-phenylenediamine sulfate (in 7.2 N HCl) and 3 mM $FeCl_3$ (in 1.2 N HCl) (added from 10x stock solutions). After 20 min at RT in the dark, the amount of methylene blue formed was determined spectrophotometrically at 670 nm. Sulfide concentrations were calculated from a standard curve prepared with $Li_2S$.

## Cryo-EM sample preparation

Samples for single-particle cryo-EM analysis were prepared under anaerobic conditions in an atmosphere of 3-5% $H_2$ in $N_2$ within an anaerobic tent (Coy Laboratories). Protein stock solutions were thawed on ice and anaerobic cryo-EM buffer (20 mM Tris-HCl pH 7.4, 100 mM NaCl, 1.1 mM NADPH) was prepared freshly. For the $(NIAUF)_2$ sample, 27.5 µM $(NIA)_2$ was mixed with 110 µM iron-loaded ISCU2 to yield the $(NIAU)_2$ complex. Subsequently, 110 µM holo-FDX2 and catalytic amounts (2.75 µM) of FDXR were added and the sample was incubated for 20 min at room temperature in cryo-EM buffer. 2.7 µL of sample was mixed with 0.3 µL of fluorinated fos-choline-8 solution (Anatrace; final concentration 1.5 mM) and applied to UltrAuFoil 0.6/1 300 mesh gold grids (Quantifoil), which had been glow-discharged two times for 90 s at 15 mA and 0,38 mbar using a PELCO easiGlow unit (Ted Pella Inc.). The sample was blotted at 4 °C and 100 % humidity and vitrified in liquid ethane using a Vitrobot Mark IV (Thermo Scientific). The $(NIAUXF)_2$ turnover sample was prepared similarly, but 110 µM FXN and 300 µM cysteine were added, prior to vitrification, resulting in a total reaction time of less than 1 min at ≤ 4 °C. We used gold support grids for our study because of their inert character and their advantage in limiting beam-induced motion during data acquisition[40].

## Data acquisition and image processing

Cryo-EM data were collected at 300 kV on a Krios G4 (Thermo Scientific) equipped with a cold field emission gun (E-CFEG) and a Falcon 4 direct electron detector with a Selectris X Imaging filter. A nominal magnification of 215,000x was used, which corresponds to a calibrated pixel size of 0.573 Å. For the $(NIAUF)_2$ sample, a dataset of 8497 EER files (952 internal frames) was collected in counting mode with a total dose of 80 e⁻/Å² using aberration-free image shift (AFIS) in EPU (Thermo Scientific) (Supplementary Table 3). For the $(NIAUXF)_2$ turnover sample, two datasets were collected from the same grid with identical settings (Supplementary Table 4); dataset 1 contained 9144 EER files (987 internal frames), dataset 2 contained 8747 EER files (1078 internal frames). These datasets were processed separately until the particle polishing step (Supplementary Fig. 5).

The overall data processing strategy until particle polishing of the three datasets was similar (Supplementary Fig. 4 and Fig. 5) and will hereafter be exemplarily described for the $(NIAUF)_2$ dataset. Movies were sorted into optics groups corresponding to their EPU AFIS metadata (https://github.com/DustinMorado/EPU_group_AFIS) before they were gain-normalized and motion-corrected with dose-weighting using MotionCor2[41] in RELION-4[42]. The contrast transfer function (CTF) was estimated with CTFFind4.1.13[43]. Particle coordinates, initially picked with the blob picker implementation in cryoSPARC live[44] during data acquisition of the respective datasets and cleaned by 2D classification, were imported into RELION and used to train a particle detection model with Topaz[45]. Topaz picking parameters were first tested and adjusted on a small particle subset before particles were picked from all micrographs and extracted using a box size of 416×416, rescaled to 104 × 104 (pixel size 2.292 Å). These particles were then imported to cryoSPARC v4.1.2[44] and subjected to 2D classification. Three ab-initio 3D reconstructions were generated from select particles, and used as templates for heterogenous refinement (3 classes, no symmetry applied). The major class (510,575 particles), revealing the characteristic appearance of the core ISC complex, was selected and subjected to homogenous refinement (no symmetry applied). The refined particle coordinates were converted into a particle STAR file using pyem[46] and imported into RELION for re-extraction using a box size of 416 × 416 rescaled to 288 × 288 (pixel size 0.828 Å), and imported to cryoSPARC for a non-uniform refinement applying C2 symmetry and optimizing per-group defocus and CTF parameters, which yielded a reconstruction at a resolution of 2.33 Å. Finally, these refined particles were imported into RELION for Bayesian particle polishing using custom parameters (trained on 10,000 particles). The polished particles were imported to cryoSPARC and subjected to a non-uniform refinement (C2 symmetry applied, per-group defocus and CTF optimization enabled), followed by a heterogenous refinement (2 classes, C2 symmetry applied). Particles from the major 3D class (363,652 particles) were subjected to a non-uniform refinement (C2 symmetry applied, per-group defocus and CTF optimization enabled), resulting in a consensus map with a global resolution of 2.03 Å. For the $(NIAUXF)_2$ turnover datasets, the global resolutions for the consensus non-uniform refinement of the two combined datasets gave a global resolution of 2.09 Å (C2 symmetry applied, per group defocus and CTF optimization enabled). The overall appearance of the two consensus maps was highly similar, but some degree of heterogeneity could be observed at the FDX2 binding site (Supplementary Figs. 1c, 2c and 3), which was further investigated by performing C2 symmetry expansion and focused 3D classification.

Particles from the $(NIAUF)_2$ consensus refinement were C2 symmetry expanded with *relion_particle_symmetry_expand* and subjected to a focused 3D classification without alignment (number of classes = 2, regularisation parameter T = 4, number of iterations = 50, initial low-pass filter = 10 Å) using a mask around the FDX2-binding region. This separated the particles into two classes; Class 1 (38.8%) shows FDX2 bound to NFS1 in the distal conformation, whereas Class 2 (61.2%) shows FDX2 bound in the groove between NFS1 and ISCU2, in the proximal conformation. Particles from both classes were imported into cryoSPARC for local refinements (no symmetry applied) yielding resolutions of 2.39 Å (Class 1) and 2.26 Å (Class 2; FDX2-bound proximal). Particles from Class 1 were imported into RELION for particle subtraction using a mask around the FDX2-binding region and subjected to a focused 3D classification without alignment (number of classes = 4, regularisation parameter T = 256, number of iterations = 50, initial low-pass filter = 10 Å). 88,788 particles from the major class (Class 3; 31.1%) were reverted to the original particles, imported to cryoSPARC and subjected to a local refinement (no symmetry applied) yielding a resolution of 2.52 Å (FDX2-bound distal).

For the $(NIAUXF)_2$ turnover dataset, polished particles from both datasets were subjected to a non-uniform refinement (C2 symmetry

applied, per group defocus and CTF optimization enabled), which converged at a resolution of 2.09 Å. The 731,041 particles from this consensus refinement were converted into a STAR file using pyem, C2 symmetry expanded with *relion_particle_symmetry_expand* and subjected to a focused 3D classification without alignment (number of classes = 3, regularisation parameter T = 16, number of iterations = 50, initial low-pass filter = 10 Å) using a mask around the FDX2-binding region. This separated the dataset into classes with FXN bound (20.4%), FDX2 bound in the proximal conformation (14.2%) and FDX2 bound in the distal conformation (65.4%). Particles from each class were imported into cryoSPARC and subjected to local refinements, reaching resolutions of 2.49 Å (FXN-bound), 2.33 Å (FDX2-bound proximal) and 2.35 Å (FDX2-bound distal). Particles of the FDX2-bound distal class were imported to RELION for particle subtraction using a mask on the FDX2-binding region before they were subjected to a focused 3D classification without alignment (number of classes = 4, regularisation parameter T = 128, number of iterations = 50, initial low-pass filter = 10 Å). The 3D class with the strongest density was selected (326,204 particles), reverted to the original particle images, imported into cryoSPARC and subjected to a local refinement (no symmetry applied), reaching a resolution of 2.46 Å.

## Model building and refinement

Half maps of the 'FDX2-bound proximal' and 'FDX2-bound distal' structures originating from the $(NIAUF)_2$ dataset were used for density modification with *phenix.resolve_cryo_em*[47] (molecular mass provided, 'real-space weighting' and 'weight by sigmas' options enabled). This improved the overall map interpretability and facilitated model building and identification of water molecules. For model refinement of the structures of the $(NIAUXF)_2$ turnover dataset, the auto-sharpened maps were used, except for the 'FXN-bound' structure, which was manually sharpened (B-factor = -60 Å$^2$).

Structures of the human $Zn^{2+}$-bound $(Zn-NIAUX)_2$ complex (PDB 6NZU)[18] and human FDX2 (PDB 2Y5C)[11] served as templates for the 'FDX2-bound proximal' and 'FDX2-bound distal' structures and were rigid-body fitted into the respective cryo-EM maps using ChimeraX[48]. The coordinates were real-space refined against the respective maps using *phenix.real_space_refine* within PHENIX[49] followed by manual fitting in Coot-0.9[50]. Metal restraints were added with *ReadySet!* in PHENIX, optimizing ligand geometry and using default bond lengths. Waters were built automatically with *phenix.douse* and checked manually for sensible hydrogen bonding. Final models were validated by *phenix.validation_cryoem*. Refinement statistics are listed in Supplementary Tables 3 and 4 for the $(NIAUF)_2$ and $(NIAUXF)_2$ turnover datasets, respectively. Map and model of cofactors are shown in Supplementary Figs. 13 and 14 for the $(NIAUF)_2$ and $(NIAUXF)_2$ turnover dataset, respectively.

## Visualization

Sequence alignments were performed with ClustalW[51] and visualized with the ESPript 3 server[52]. All maps and models were visualized in ChimeraX[48]. The electrostatic potential map in Fig. 2d was calculated with the APBS-PDB2PQR software suite[53] and displayed in ChimeraX.

## Reporting summary

Further information on research design is available in the Nature Portfolio Reporting Summary linked to this article.

## Data availability

Cryo-EM maps and atomic models from the $(NIAUF)_2$ dataset were deposited to the Electron Microscopy Data Bank and the Protein Data Bank under the accession codes EMD-19355 ($(NIAUF)_2$ consensus map), EMD-19356 and PDB 8RMC (FDX2-bound proximal), EMD-19357 and PDB 8RMD (FDX2-bound distal). Cryo-EM maps and atomic models originating from the $(NIAUXF)_2$ turnover datasets are accessible under EMD-19358 ($(NIAUXF)_2$ turnover, consensus map), EMD-19359 and PDB 8RME ($(NIAUXF)_2$ turnover, FXN-bound), EMD-19360 and PDB 8RMF ($(NIAUXF)_2$ turnover, FDX2-bound proximal), EMD-19361 and PDB 8RMG ($(NIAUXF)_2$ turnover, FDX2-bound distal). Source data are provided with this paper.

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

## Acknowledgements

We thank the Central Electron Microscopy Facility of the Max Planck Institute of Biophysics for providing cryo-EM infrastructure and technical support, and Werner Kühlbrandt and the Department of Structural Biology for support and helpful discussions. We acknowledge the contribution of the Core Facility 'Protein Biochemistry and Spectroscopy' of the Philipps-Universität Marburg. This work was funded by the Max Planck Society (to B.J.M.) and by generous financial support from Deutsche Forschungsgemeinschaft (SPP 1927, LI 415/7 to R.L.).

## Author contributions

R.S. prepared cryo-EM samples, acquired and processed cryo-EM data, built and refined the atomic models, analyzed data, drew figures, wrote the initial draft and edited the manuscript. L.B. purified proteins, performed and analyzed FeS reconstitution assays and titrations, prepared figures, wrote and edited the manuscript. S.A.F. purified proteins, performed and analyzed the desulfurase assay and revised the manuscript. V.S. purified proteins, performed and analyzed the dithionite assay and revised the manuscript. N.K. purified proteins, performed and analyzed the desulfurase assay and revised the manuscript. S.K. supported cryo-EM data acquisition and revised the manuscript. R.L. and B.J.M. initiated the study and supervised the project, analyzed data, wrote and edited the manuscript.

## Funding

## Competing interests

The authors declare no competing interests.
