## [Transparent Peer Review file · Nature Communications]

Two-stage binding of mitochondrial ferredoxin-2 to the core iron-sulfur cluster assembly complex

Corresponding Author: Dr Bonnie Murphy

Version 0:

Reviewer comments:

Reviewer #1

(Remarks to the Author)

Iron-sulfur (Fe-S) cluster biosynthesis is a fundamental process across all domains of life, crucial for the proper assembly and integration of Fe-S clusters into target proteins. In eukaryotes, the initial step of Fe-S cluster assembly is orchestrated by the mitochondrial iron-sulfur cluster assembly (ISC) system containing the scaffold protein ISCU2, the cysteine desulfurase subcomplex NFS1-ISD11-ACP1, the allosteric activator frataxin (FXN) and the electron donor ferredoxin 2 (FDX2). Despite the extensive studies, the mechanism of electron transfer from FDX2 to the complex remains elusive. By using cryoEM, Ralf et al. uncover two distinct conformations of FDX2 binding to the core human ISC complex. The manuscript presents new and exciting results, shedding light on the structural basis for a plausible mechanism of electron transfer from FDX2 to the core ISC complex and the coordinated interactions of FDX2 and FXN with the core ISC complex during the de novo Fe-S cluster biosystem event.

a notable concern raised by this reviewer pertains to the Fe coordination in ISCU2 in the presence of FDX2 or FXN presented in Figure 4. The proposed presence of a bidentate Asp at the four-coordinate Fe site when FDX2 (or FXN) is bound in the distal conformation appears highly unusual, if not unprecedented. Detailed coordination information, including bond lengths and angles of the Fe center, is warranted for proper evaluation of the Fe binding site. What does the density map look like around the region? Is there any unexplained electron density near the Fe center to suggest an alternative coordination environment? The authors should cite the literature, including such a Fe binding site, if any, and consider employing additional spectroscopic techniques (such as XAS, Mössbauer, or EPR) to validate the proposed structure. At the very least, the authors should emphasize the uncertainty and limitations of the study to the readers.

Other comments and suggestions for consideration:

1) Line 26: "...C terminus forms contacts with NFS1; in this conformation..."

Change "; in" to ". In"

2) Line 120: "namely 23 Å for the distal and 14 Å for the proximal conformation (Fig. 2A,B)."

Add "conformation" after "distal"

3) Line 362: "...conformation, whereas Class2..."

Change to "Class 2"

4) Figure 1: Consider improving the color scheme and presentation style for better data interpretation.

Reviewer #2

(Remarks to the Author)

In this manuscript, Steinhilper et al. present cryo-EM reconstructions of the human core ISC in complex with FDX2. They show that FDX2 binds in two different conformations, termed proximal and distal, to the core ISC, and that FDX2 interacts with an Arg-patch on NFS1. In the proximal conformation, the flexible C-terminus of FDX2 interacts stably with NFS1. In this conformation, the distance between the 2Fe-2S cluster of FDX2 and the FeS cluster assembly site of ISCU2 is reduced from 23 Å to 14 Å, suitable for e- transfer.

Since the initial goal of this study had been to determine a reconstruction of ISC bound to both FXN and FDX2, the authors also prepared a turnover sample that contained ISC, FDX2 and FXN, as well as Cys and FDXR. From this sample, the authors extracted three different states: ISC bound to FDX2 in the proximal and distal conformations (as observed in the

(NIAUXF)2 complex before), as well as ISC bound to FXN. The authors conclude that FXN and FDX2 bind to overlapping binding sites, and that FXN and FDX2 cannot bind simultaneously to the ISC. This is in contrast to the model proposed previously by Boniecki et al., 2017, which was based on low resolution SAXS data of a *C. thermophilum* complex; binding of FXN is in agreement with a published cryo-EM structure of a frataxin bound complex (Fox et al., 2019).

The structural data presented in this manuscript is of very high quality. By using anaerobic conditions for both sample assembly and grid preparation, the authors ensure that the sample is in a physiologically relevant state. This clearly sets it apart from previous structural studies, which were done under aerobic conditions and with Zn bound instead of Fe (Fox et al., 2019). Through careful data processing, the authors succeeded in extracting different states from the cryo-EM data sets, and to obtain reconstructions at near-atomic resolution for all states.

While the structures containing FDX2 are novel, the structure containing FXN is “virtually identical” (p. 5) with the previously published ones, not offering major novel insights by itself.

While the structural data is well done and convincing, some of the structural insights would benefit from functional and mechanistic validation *in vitro*, e.g.

- Validation of the binding interfaces through mutational analysis or biophysical binding studies, determination of dissociation constants -> this might also offer valuable insights regarding how binding of FXN vs. FDX2 is regulated
- Is the C-terminal Lys-Pro-His motif dispensable for binding to the ISC? It was unexpected that it is disordered upon binding, so this would be interesting to test biochemically

It would also be helpful to discuss how the data compares to the study by Cai et al., *Biochemistry*, 2017, which characterized FDX1 and FDX2 (in oxidized and reduced states) binding to ISC core complex by NMR.

In the discussion section, it would be interesting to consider how the binding of FXN vs. FDX2 might be regulated. Is one favored over the other at different steps of biogenesis, and why? Are the cellular concentrations different?

Minor comments and questions:

- It should be mentioned explicitly in the abstract and methods that the sequences for human proteins were used
- What was the rationale for using NADPH-FDXR-reduced FDX2, instead of treatment with sodium dithionite (as was done in previous work by Boniecki et al., 2017)?
- In the turnover sample, the authors identified three different complexes, FXN-bound, and FDX2-bound distal and proximal. Are there other states within that sample, e.g. corresponding to step 6 in Figure 5 (dissociated FDX2, ISCU2 dimer)?
- In figure 3, it would be helpful to have a zoomed-out view to clarify where exactly on NFS1 the C-terminus binds; it might also be helpful to include this in the movie (we understand that it was omitted for clarity, but given the importance of this loop, it would be helpful for the reader to be able to visualize where exactly it is bound)
- In figure 4, it would be helpful to label FDX2 as FDX2 (distal) and FDX2 (proximal) so that this is obvious at first glance
- In Movie S1, it would be helpful to label helix F, and highlight the Cys in Cys-loop of ISCU2
- In movie S1, while morphing, there seems to be a superfluous rotation in the representation of the ISCU2 loop to beta sheet transition (@19', 20', 32', 35')

Reviewer #3

(Remarks to the Author)

The manuscript by Steinhilper, et al describes a new cryo-EM structure of the human core ISC complex that reveal new information about binding modes for the ferredoxin FDX2. The paper is very well written and was enjoyable to read. I also appreciated the effort the authors put into the design of their structural figures. The manuscript consists of descriptions and insights from several cryo-EM structures, so there is not a tremendous amount of experimental detail or data analysis to comment on. The structures clarify that FXN and FDX2 binding are mutually exclusive and support the proposed model in Figure 5. I think the structural identification of a potential two-step binding process for FDX2 to the ISC complex is also an important result. I'm curious if there is any biophysical evidence that supports the proposed mechanism? For example, the interaction between the SufS/SufE cysteine desulfurase/transpersulfurase pair is predicted to use a two-step binding model supported by a ten-fold improvement in Kd values under various activating conditions (JBC, 2013, 288, 36189 and JBC, 2023, 299, 102966). Other than this minor question, I support publication of the current manuscript version.

Reviewer #4

(Remarks to the Author)

I co-reviewed this manuscript with one of the reviewers who provided the listed reports. This is part of the Nature Communications initiative to facilitate training in peer review and to provide appropriate recognition for Early Career Researchers who co-review manuscripts

Version 1:

Reviewer comments:

Reviewer #1

(Remarks to the Author)

The authors have addressed most of my questions.

Reviewer #2

(Remarks to the Author)

The authors have expanded their manuscript significantly by adding new experimental data to functionally validate the interactions identified in their structural models. The authors have addressed my comments, and the manuscript now presents as a well-rounded study which provides significant novel insights into the role of FDX2.

Reviewer #3

(Remarks to the Author)

The addition of the in vitro characterization of the binding surfaces for the FDX2 with the ISC complex serves to enhance the impact of the new structures. The authors make a good point that the shift from distal to proximal does not require an exogenous trigger, which satisfies my main mechanistic concern. The authors provide a reasonable hypothesis for regulation of the FXN/FDX2 choreography. I support publication of the revised manuscript.

Reviewer #4

(Remarks to the Author)

REVIEWER COMMENTS

Reviewer #1 (Remarks to the Author):

Iron-sulfur (Fe-S) cluster biosynthesis is a fundamental process across all domains of life, crucial for the proper assembly and integration of Fe-S clusters into target proteins. In eukaryotes, the initial step of Fe-S cluster assembly is orchestrated by the mitochondrial iron-sulfur cluster assembly (ISC) system containing the scaffold protein ISCU2, the cysteine desulfurase subcomplex NFS1-ISD11-ACP1, the allosteric activator frataxin (FXN) and the electron donor ferredoxin 2 (FDX2). Despite the extensive studies, the mechanism of electron transfer from FDX2 to the complex remains elusive. By using cryoEM, Ralf et al. uncover two distinct conformations of FDX2 binding to the core human ISC complex. The manuscript presents new and exciting results, shedding light on the structural basis for a plausible mechanism of electron transfer from FDX2 to the core ISC complex and the coordinated interactions of FDX2 and FXN with the core ISC complex during the de novo Fe-S cluster biosystem event.

a notable concern raised by this reviewer pertains to the Fe coordination in ISCU2 in the presence of FDX2 or FXN presented in Figure 4. The proposed presence of a bidentate Asp at the four-coordinate Fe site when FDX2 (or FXN) is bound in the distal conformation appears highly unusual, if not unprecedented. Detailed coordination information, including bond lengths and angles of the Fe center, is warranted for proper evaluation of the Fe binding site. What does the density map look like around the region? Is there any unexplained electron density near the Fe center to suggest an alternative coordination environment? The authors should cite the literature, including such a Fe binding site, if any, and consider employing additional spectroscopic techniques (such as XAS, Mössbauer, or EPR) to validate the proposed structure. At the very least, the authors should emphasize the uncertainty and limitations of the study to the readers.

We thank the reviewer for this point. We agree that this coordination geometry is unusual. However, we note that monodentate ligation by Asp71 gives three-coordinate Fe, which would also be unusual. The local interpretability at the Fe site is lower, likely due to conformational plasticity of this site. Therefore, it is entirely possible that additional (solvent) ligands to the Fe are unresolved in our map. Spectroscopic investigations of the metal coordination are complicated by, e.g., the mixture of ISC complex states due to the competitive binding of FDX2 and FXN binding. The studies would certainly require the involvement of additional labs (experts in Mössbauer or EPR spectroscopy). For these reasons, we consider such studies beyond the scope of the paper, which does not particularly address the coordination at the Fe site.

We have added the following text to the paper to emphasize to the reader the uncertainty of this assignment:

'We note that poorer local resolution at the Fe-binding site, likely caused by its structural malleability, means that additional (solvent) ligands to the Fe site may not be resolved.'

Other comments and suggestions for consideration:

1) Line 26: "...C terminus forms contacts with NFS1; in this conformation..."
Change "; in" to ". In"

Done, thank you.

2) Line 120: "namely 23 Å for the distal and 14 Å for the proximal conformation (Fig. 2A,B)."
Add "conformation" after "distal"

Done, thank you.

3) Line 362: "...conformation, whereas Class2...."
Change to "Class 2"

Done, thank you.

4) Figure 1: Consider improving the color scheme and presentation style for better data interpretation.

We have looked carefully at the presentation style and color scheme of figure 1 but are unsure what changes would ease its interpretation. The color scheme is chosen to provide contrast between adjacent subunits while intuitively marking out the two copies of each subunit in the NIAU dimer. We have checked that colleagues with red-green color blindness can easily distinguish the colors chosen. We remain open to requests for specific changes from the side of the reviewer(s) and editor.

Reviewer #2 (Remarks to the Author):

In this manuscript, Steinhilper et al. present cryo-EM reconstructions of the human core ISC in complex with FDX2. They show that FDX2 binds in two different conformations, termed proximal and distal, to the core ISC, and that FDX2 interacts with an Arg-patch on NFS1. In the proximal conformation, the flexible C-terminus of FDX2 interacts stably with NFS1. In this conformation, the distance between the 2Fe-2S cluster of FDX2 and the FeS cluster assembly site of ISCU2 is reduced from 23 Å to 14 Å, suitable for e- transfer.

Since the initial goal of this study had been to determine a reconstruction of ISC bound to both FXN and FDX2, the authors also prepared a turnover sample that contained ISC, FDX2 and FXN, as well as Cys and FDXR. From this sample, the authors extracted three different states: ISC bound to FDX2 in the proximal and distal conformations (as observed in the (NIAUXF)₂ complex before), as well as ISC bound to FXN. The authors conclude that FXN and FDX2 bind to overlapping binding sites, and that FXN and FDX2 cannot bind simultaneously to the ISC. This is in contrast to the model proposed previously by Boniecki et al., 2017, which was based on low resolution SAXS data of a *C. thermophilum* complex; binding of FXN is in agreement with a published cryo-EM structure of a frataxin bound complex (Fox et al., 2019).

The structural data presented in this manuscript is of very high quality. By using anaerobic conditions for both sample assembly and grid preparation, the authors ensure that the sample is in a physiologically relevant state. This clearly sets it apart from previous structural studies, which were done under aerobic conditions and with Zn bound instead of Fe (Fox et al., 2019). Through careful data processing, the authors succeeded in extracting different states from the cryo-EM data sets, and to obtain reconstructions at near-atomic resolution for all states.

While the structures containing FDX2 are novel, the structure containing FXN is "virtually identical" (p. 5) with the previously published ones, not offering major novel insights by itself.

While the structural data is well done and convincing, some of the structural insights would benefit from functional and mechanistic validation in vitro, e.g.

- Validation of the binding interfaces through mutational analysis or biophysical binding studies, determination of dissociation constants -> this might also offer valuable insights regarding how binding of FXN vs. FDX2 is regulated

We have extended the manuscript significantly to comprehensively address the biochemical validation of the structures by testing mutant proteins in a functional approach. First, we have generated mutant FDX2 (and NFS1) proteins carrying alterations in all three identified contact areas between FDX2 and NFS1 or ISCU2 within the (Fe-NIAUF)₂ complexes. These areas encompass i) the salt bridges formed by helix F of FDX2 with an Arg-rich patch of NFS1 in both the distal and proximal conformations, ii) the C terminus of FDX2 that associates with NFS1 in the proximal conformation only, and iii) a potential interaction site between FDX2 and ISCU2. We then functionally evaluated these variants in extensive biochemical analyses using a CD-based [2Fe-2S] cluster biosynthesis assay established earlier in our lab (Webert et al., 2014). In carrying out these experiments, we found that, for the human proteins, the effect of FDX2 concentration on the cluster synthesis rate exhibits a bell-shaped dependence. A similar competitive behavior was observed upon varying the FXN concentration in this synthesis assay. Likely, these results reflect the competition between FDX2 and FXN for an overlapping binding site (as seen in our structures and biochemically observed in yeast (Uzarska et al., 2022)). In a nutshell, our results presented in the new Figs. 5 and 6 and new Suppl. Figs. 8-12 identify residues and regions of importance for functionality of FDX2 in de novo [2Fe-2S] cluster assembly. For instance, we show that the salt bridge between FDX2 and NFS1 is important for optimal cluster synthesis rates. In spite of its role in determining the human ferredoxin isoform specificity (Schulz et al., Nature Chem. Biol 2023), the conserved C terminus of FDX2 was not found to be essential for its in vitro function. However, C-terminal FDX2 truncations and a site-directed mutant FDX2 point to the functional importance of Asn175, which forms hydrogen bonds with Ser385 of NFS1 in our structure. This interaction may facilitate the movement of FDX2 from the distal to the proximal conformation. Finally, the mild effect observed by mutation of a potential interaction area of FDX2 with ISCU2 and the lack of possible specific contacts within the proximal (Fe-NIAUF)₂ structure makes it unlikely that this area is of functional importance.

- Is the C-terminal Lys-Pro-His motif dispensable for binding to the ISC? It was unexpected that it is disordered upon binding, so this would be interesting to test biochemically

The C-terminal Lys-Pro-His motif was part of the studies discussed in the previous paragraph.

It would also be helpful to discuss how the data compares to the study by Cai et al., Biochemistry, 2017, which characterized FDX1 and FDX2 (in oxidized and reduced states) binding to ISC core complex by NMR.

An intensive comparison to the study by Cai et al. has been made in our previous publication evaluating the distinct functions of FDX1 and FDX2 (Schulz et al., Nature Chem. Biol 2023), and the Reviewer is referred to this study. We show in this previous study (confirming earlier in vivo data by Sheftel et al. PNAS 2010) that FDX1 has no function in Fe/S protein biogenesis. The differences to the study by Cai et al. may be explained by their use of dithionite for FDX1/2 reduction in [2Fe-2S] cluster synthesis. As addressed below, DT-mediated reduction as compared to NADPH-FDXR-assisted reduction (typically used here) may alter the properties of Fe/S cluster-containing biomolecules and may generate artificial products.

In the discussion section, it would be interesting to consider how the binding of FXN vs. FDX2 might be regulated. Is one favored over the other at different steps of biogenesis, and why? Are the cellular concentrations different?

This is indeed an interesting problem. Here, we show that FDX2 and FXN compete for an overlapping binding site, and optimal concentrations are needed for maximum synthesis rates. We also show that lower concentrations of FDX2 than of FXN are needed for optimal cluster synthesis rates, suggesting that FXN binds more weakly to the binding site in (NIAU)₂

(Figure 5e, 5h, 5i). We have added a reference to quantitative proteomics data suggesting that the mitochondrial levels of FDX2 are nearly 5-fold lower than those of FXN in HEK293T cells (Morgenstern et al., Cell Metabolism 2021). Future dedicated studies would be needed to answer the interesting question of whether ISC protein concentration variations are used physiologically for regulatory purposes.

Minor comments and questions:

- It should be mentioned explicitly in the abstract and methods that the sequences for human proteins were used

Done.

- What was the rationale for using NADPH-FDXR-reduced FDX2, instead of treatment with sodium dithionite (as was done in previous work by Boniecki et al., 2017)?

We have used NADPH-FDXR reduction for functional studies ([2Fe-2S] cluster synthesis in the CD-based assay, also in Boniecki et al. 2017) since 2014 (Webert et al.), because this reduction system closely replicates in vivo conditions. In the SAXS analysis shown in Boniecki et al. 2017, the use of dithionite (DT) was technically necessary to avoid interference with other proteins in these low-resolution structural studies. The use of DT as a reductant should, however, be restricted to systems where NADPH-FDXR cannot be used. As a strong chemical reductant, DT may alter cluster binding properties or interfere with reduction-sensitive proteins in a non-physiological manner. An example is presented in Fig. 6c-e and Supplementary figure 12 for the FDX2-DA variant that cannot bind FDXR and hence requires reduction by DT. As we show, the [2Fe-2S] cluster produced by DT-reduced FDX2 slightly differs from the product generated by physiologically reduced FDX2. Hence, experiments using DT have to be interpreted with caution.

- In the turnover sample, the authors identified three different complexes, FXN-bound, and FDX2-bound distal and proximal. Are there other states within that sample, e.g. corresponding to step 6 in Figure 5 (dissociated FDX2, ISCU2 dimer)?

We did not resolve any additional structures, such as an ISCU2 dimer (too small to be resolved by cryo-EM). We emphasize that we do not rule out the possibility that such complexes might be present in the sample. It is possible that short-lived (underpopulated) states are not detected. Initial 2D classes of the particles extracted from the dataset after Topaz picking are attached (for review only) and show no indication of such a complex.

- In figure 3, it would be helpful to have a zoomed-out view to clarify where exactly on NFS1 the C-terminus binds; it might also be helpful to include this in the movie (we understand that it was omitted for clarity, but given the importance of this loop, it would be helpful for the reader to be able to visualize where exactly it is bound)

Done, thank you.

- In figure 4, it would be helpful to label FDX2 as FDX2 (distal) and FDX2 (proximal) so that this is obvious at first glance

Done, thank you.

- In Movie S1, it would be helpful to label helix F, and highlight the Cys in Cys-loop of ISCU2

Helix F and ISCU2 FeS assembly site are now labelled in Movie S1. Moreover, the C terminus has been added for the proximal conformation of the (Fe-NIAUF)₂ complex.

- In movie S1, while morphing, there seems to be a superfluous rotation in the representation

of the ISCU2 loop to beta sheet transition (@19', 20', 32', 35')

Corrected, thank you.

Reviewer #3 (Remarks to the Author):

The manuscript by Steinhilper, et al describes a new cryo-EM structure of the human core ISC complex that reveal new information about binding modes for the ferredoxin FDX2. The paper is very well written and was enjoyable to read. I also appreciated the effort the authors put into the design of their structural figures. The manuscript consists of descriptions and insights from several cryo-EM structures, so there is not a tremendous amount of experimental detail or data analysis to comment on. The structures clarify that FXN and FDX2 binding are mutually exclusive and support the proposed model in Figure 5. I think the structural identification of a potential two-step binding process for FDX2 to the ISC complex is also an important result. I'm curious if there is any biophysical evidence that supports the proposed mechanism? For example, the interaction between the SufS/SufE cysteine desulfurase/transpersulfurase pair is predicted to use a two-step binding model supported by a ten-fold improvement in K_d values under various activating conditions (JBC, 2013, 288, 36189 and JBC, 2023, 299, 102966). Other than this minor question, I support publication of the current manuscript version.

We appreciate this positive evaluation. As mentioned in detail for Reviewer 2, we have added a substantial amount of biochemical data for mutant FDX2 and NFS1 proteins addressing the FDX2 binding regions at the core ISC complex. These results are closely related to the concerns raised by this Reviewer. The comparison of the ISC system to the SUF system is certainly an interesting question. The prediction of two binding states in the SufS/SufE system was based on distinguishing states +/- added Cys and SufS mutant analyses (if this is what the reviewer refers to). This system differs from our situation where we have two consecutive binding steps without further additions. Distinguishing the distal and proximal FDX2 binding steps would require, e.g., fast (stopped flow?) binding kinetics, a complex analysis that we believe is not in the scope of this manuscript. We would like to mention that several characteristics distinguish the SUF and ISC systems and make a simple comparison difficult. SUF uses another scaffold (SufBCD), and therefore only the sulfur release and transfer reactions may be relevant for comparison. NFS1 (or bacterial IscS) and SufS are distinguished by an extra sequence patch and thus characteristically differ in structure. Therefore, even the sulfur release step characteristically differs for NFS1 and SufS. Even more so, sulfur transfer from SufS to SufU (or SufE) differs from the ISC system in the important aspect that Cys1 in the SUF system serves as the sulfur acceptor, while Cys3 (i.e. the third of three conserved Cys) is used in the ISC system. This is also evident from structural comparisons of the (NIAU)2 and SufSU complexes. We therefore believe that these comparative aspects do not fit into the current study. We further note that aspects of sulfur transfer from NFS1 to ISCU2 have been discussed and briefly compared to SUF in our previous paper (Schulz et al. 2024).

Reviewer #4 (Remarks to the Author):

We also thank Reviewer #4.